# Stop codon context influences genome-wide stimulation of termination codon readthrough by aminoglycosides

**Jamie R Wangen, Rachel Green***

Department of Molecular Biology and Genetics, Howard Hughes Medical Institute, Johns Hopkins University School of Medicine, Baltimore, United States

**Abstract** Stop codon readthrough (SCR) occurs when the ribosome miscodes at a stop codon. Such readthrough events can be therapeutically desirable when a premature termination codon (PTC) is found in a critical gene. To study SCR in vivo in a genome-wide manner, we treated mammalian cells with aminoglycosides and performed ribosome profiling. We find that in addition to stimulating readthrough of PTCs, aminoglycosides stimulate readthrough of normal termination codons (NTCs) genome-wide. Stop codon identity, the nucleotide following the stop codon, and the surrounding mRNA sequence context all influence the likelihood of SCR. In comparison to NTCs, downstream stop codons in 3′UTRs are recognized less efficiently by ribosomes, suggesting that targeting of critical stop codons for readthrough may be achievable without general disruption of translation termination. Finally, we find that G418-induced miscoding alters gene expression with substantial effects on translation of histone genes, selenoprotein genes, and *S*-adenosylmethionine decarboxylase (AMD1).

*For correspondence:
ragreen@jhmi.edu

## Introduction

To complete synthesis of a mature protein, ribosomes must terminate translation accurately at the end of each coding sequence. Translation termination requires recognition of the normal termination codon (NTC) in the A site of the ribosome by release factors eRF1 and eRF3 in eukaryotes, a process distinct from the RNA-RNA mediated decoding of mRNAs by tRNAs during translation elongation (*Schuller and Green, 2018*). Following stop codon recognition by release factors, the nascent peptide is hydrolyzed by eRF1 releasing the mature protein product (*Frolova et al., 1999*; *Zhouravleva et al., 1995*) and ribosomes are subsequently removed from the mRNA by recycling (*Barthelme et al., 2011*; *Pisarev et al., 2010*; *Shoemaker and Green, 2011*). Normally, termination is a highly efficient process, with an estimated accuracy of greater than 99% (*Floquet et al., 2012*; *Harrell et al., 2002*; *Namy et al., 2001*) ensuring maintenance of proteome fidelity.

Nonsense mutations perturb the process of translation by insertion of a premature termination codon (PTC) within the coding sequence (CDS) of a gene. Upon encountering a PTC, a ribosome will terminate translation prematurely resulting in the production of a truncated peptide, which typically lacks proper functionality, and may even exert dominant-negative effects (*Miller and Pearce, 2014*). To guard against the negative consequences of mRNAs harboring PTCs, the conserved nonsense-mediated decay (NMD) machinery, working together with the ribosome, identifies premature stop codons and targets the message for decay (*Celik et al., 2015*; *Kim and Maquat, 2019*). Discrimination of normal and problematic termination contexts must occur independently of the nucleotide sequence of the stop codon since identical stop codons (UAA, UAG, and UGA) signal translation termination at both NTCs and PTCs. In mammals, a strong signal for NMD derives from the position of the stop codon relative to that of a protein complex known as the Exon-Junction-Complex (EJC) deposited upstream of each splice junction during splicing of the mRNA (*Le Hir, 2000*; *Singh et al.,*

**eLife digest** Many genes provide a set of instructions needed to build a protein, which are read by structures called ribosomes through a process called translation. The genetic information contains a short, coded instruction called a stop codon which marks the end of the protein. When a ribosome finds a stop codon it should stop building and release the protein it has made.

Ribosomes do not always stop at stop codons. Certain chemicals can actually prevent ribosomes from detecting stop codons correctly, and aminoglycosides are drugs that have exactly this effect. Aminoglycosides can be used as antibiotics at low doses because they interfere with ribosomes in bacteria, but at higher doses they can also prevent ribosomes from detecting stop codons in human cells. When ribosomes do not stop at a stop codon this is called readthrough. There are different types of stop codons and some are naturally more effective at stopping ribosomes than others.

Wangen and Green have now examined the effect of an aminoglycoside called G418 on ribosomes in human cells grown in the laboratory. The results showed how ribosomes interacted with genetic information and revealed that certain stop codons are more affected by G418 than others. The stop codon and other genetic sequences around it affect the likelihood of readthrough. Wangen and Green also showed that sequences that encourage translation to stop are more common in the area around stop codons.

These findings highlight an evolutionary pressure driving more genes to develop strong stop codons that resist readthrough. Despite this, some are still more affected by drugs like G418 than others. Some genetic conditions, like cystic fibrosis, result from incorrect stop codons in genes. Drugs that promote readthrough specifically in these genes could be useful new treatments.

*2012*). While NTCs are typically found in the terminal exon of protein coding genes, PTCs often are found in upstream exons, and in these cases are recognized as aberrant when the ribosome encounters a termination codon upstream of a deposited EJC.

As nonsense mutations account for approximately 11% of inherited genetic disorders in humans (*Mort et al., 2008*), targeted treatments for these particular mutations could substantially alleviate human disease. One class of compound proposed as a treatment for nonsense mutations are collectively known as nonsense-suppression therapeutics (*Keeling et al., 2014*; *Lee and Dougherty, 2012*). Acting at the level of translation, compounds in this class force the ribosome to 'read through' a PTC and continue translation thereby restoring synthesis of full-length protein. Typically, such stop codon readthrough (SCR) involves a process in which a near-cognate tRNA (nc-tRNA) base pairs with a termination codon, forcing the ribosome to continue elongation instead of terminating translation (*Brody and Yanofsky, 1963*; *Smith et al., 1966*). Achieving such specificity for PTC readthrough without globally disrupting termination at NTCs remains a critical challenge for nonsense suppression therapies and will require a more complete understanding of translation termination and stop codon readthrough in different sequence contexts.

While translation termination is generally the predominant reaction at stop codons, termination efficiencies do vary considerably between different stop codon contexts. Many factors have been reported to influence the probability of termination, readthrough, or frameshifting (where the ribosome slides on an mRNA, changing the frame of translation) including the identity of the stop codon and surrounding sequence contexts (*Anzalone et al., 2019*; *Bonetti et al., 1995*; *Floquet et al., 2012*; *Harrell et al., 2002*; *McCaughan et al., 1995*; *Namy et al., 2001*), proximal RNA structures (*Firth et al., 2011*; *Steneberg and Samakovlis, 2001*), RNA modifications (*Karijolich and Yu, 2011*), presence of RNA binding proteins (*Amrani et al., 2004*), and availability of aminoacylated nc-tRNA (*Beznosková et al., 2019*; *Blanchet et al., 2015*; *Roy et al., 2015*). Intriguingly, high rates of SCR have been documented in diverse organisms including viruses (*Li and Rice, 1993*; *Wills et al., 1991*), yeast (*Namy et al., 2003*; *Williams et al., 2004*), flies (*Dunn et al., 2013*), and humans (*Loughran et al., 2014*; *Loughran et al., 2018*). Many sequences promoting readthrough show evolutionary conservation suggesting functional importance (*Jungreis et al., 2011*). Some organisms even differentially decode all three stop-codons as sense or stop in a manner thought to depend on proximity to the polyA tail (*Heaphy et al., 2016*; *Swart et al., 2016*; *Záhonová et al., 2016*). Numerous documented examples of substantial SCR (on the order of 15%) demonstrate the

potential for restoring significant levels of full-length protein in the context of nonsense mutations. Despite extensive study, comprehensive understanding of the rules dictating readthrough efficiencies of stop codons has remained elusive.

To explore the processes of translation termination and stop codon readthrough, we utilized a class of compounds known as aminoglycosides (AG). AGs have been abundantly characterized in bacterial systems where they bind the decoding center of the ribosome (*Carter et al., 2000*; *Fourmy et al., 1996*; *Moazed and Noller, 1987*) and promote miscoding (*Davies and Davis, 1968*; *Pape et al., 2000*). Similarly, in eukaryotic systems, a subset of the AGs have been shown to promote miscoding (*Palmer et al., 1979*) by binding in the decoding center of the ribosome (*Garreau de Loubresse et al., 2014*; *Prokhorova et al., 2017*). Again, in these systems, the manner in which AGs increase the likelihood of SCR is by promoting the accommodation of nc-tRNAs (*Burke and Mogg, 1985*; *Howard et al., 1996*; *Manuvakhova et al., 2000*), and simultaneously blocking the action of termination factors (*Eyler and Green, 2011*). While a correlation between basal and AG-stimulated SCR frequencies has been reported for a collection of stop codon contexts (*Floquet et al., 2012*), the extent of global perturbations on translation termination by AGs has not been documented.

Here, we performed ribosome profiling (*Ingolia et al., 2011*) to systematically investigate the activities of AGs in promoting readthrough of stop codons genome-wide. This approach allows for an unbiased examination of translation termination for all stop codons in their native sequence contexts. We observe broad stimulation of SCR following treatment with AGs that is especially robust for G418. Using these compounds, we uncover a general role for termination codon identity, as well as surrounding sequence contexts, in determining genome-wide rates of SCR. G418 stimulates readthrough at multiple classes of stop codons including PTCs, NTCs, and 3′UTR termination codons (3′TCs). Importantly, we find that G418 more potently induces readthrough of 3′TCs relative to NTCs. Finally, we define several biological processes that are disrupted by high levels of G418-induced stop codon readthrough including translation of histone mRNAs, selenoproteins, and *S*-adenosylmethionine decarboxylase 1 (AMD1).

## Results

### Aminoglycosides promote PTC readthrough while simultaneously inhibiting normal protein synthesis

We initially investigated the ability of AGs to promote readthrough of a PTC using a dual-luciferase reporter assay. To directly compare activity of various AGs, and additionally optimize AG concentrations for downstream ribosome profiling experiments, we developed a reporter expressing Firefly (FLuc) and Nano Luciferase (NLuc) from a bidirectional CMV promoter (*Figure 1A*). Normal protein synthesis was measured by the expression of full-length FLuc, and SCR was measured by the expression of NLuc when a PTC was inserted at position R154X (UGA). While expressing each luciferase from a separate mRNA can introduce variability in RNA levels between messages, this strategy avoids many of the pitfalls that can confound interpretation of bicistronic reporters for studying stop codon readthrough (*Loughran et al., 2017*; *Terenin et al., 2017*).

Stimulation of ribosome readthrough was tested for six different AGs (depicted in *Figure 1—figure supplement 1*): G418, gentamicin, paromomycin, neomycin, tobramycin, and amikacin. To compare induction of ribosome readthrough by these AGs, HEK293T cells were transiently transfected with the reporter, treated with AGs, and measured 24 hr later for luciferase activity. Consistent with the role of AGs as general protein synthesis inhibitors (*Blanchard et al., 2010*), production of FLuc decreased with increasing concentrations of AGs (*Figure 1B*). Despite this reduction in FLuc synthesis, AGs promoted NLuc synthesis on the R154X reporter, revealing strong stimulation of PTC readthrough (*Figure 1C*). When we calculated the level of normalized readthrough by normalizing the ratio of NLuc to FLuc for AG-treated cells relative to untreated cells (*Figure 1D*), all AGs tested, with the exception of tobramycin, were able to stimulate readthrough of the R154X PTC. Of the six AGs examined, G418 and gentamicin potently stimulated PTC readthrough in this assay, while paromomycin, neomycin, and amikacin showed lower but detectable stimulation of PTC readthrough.

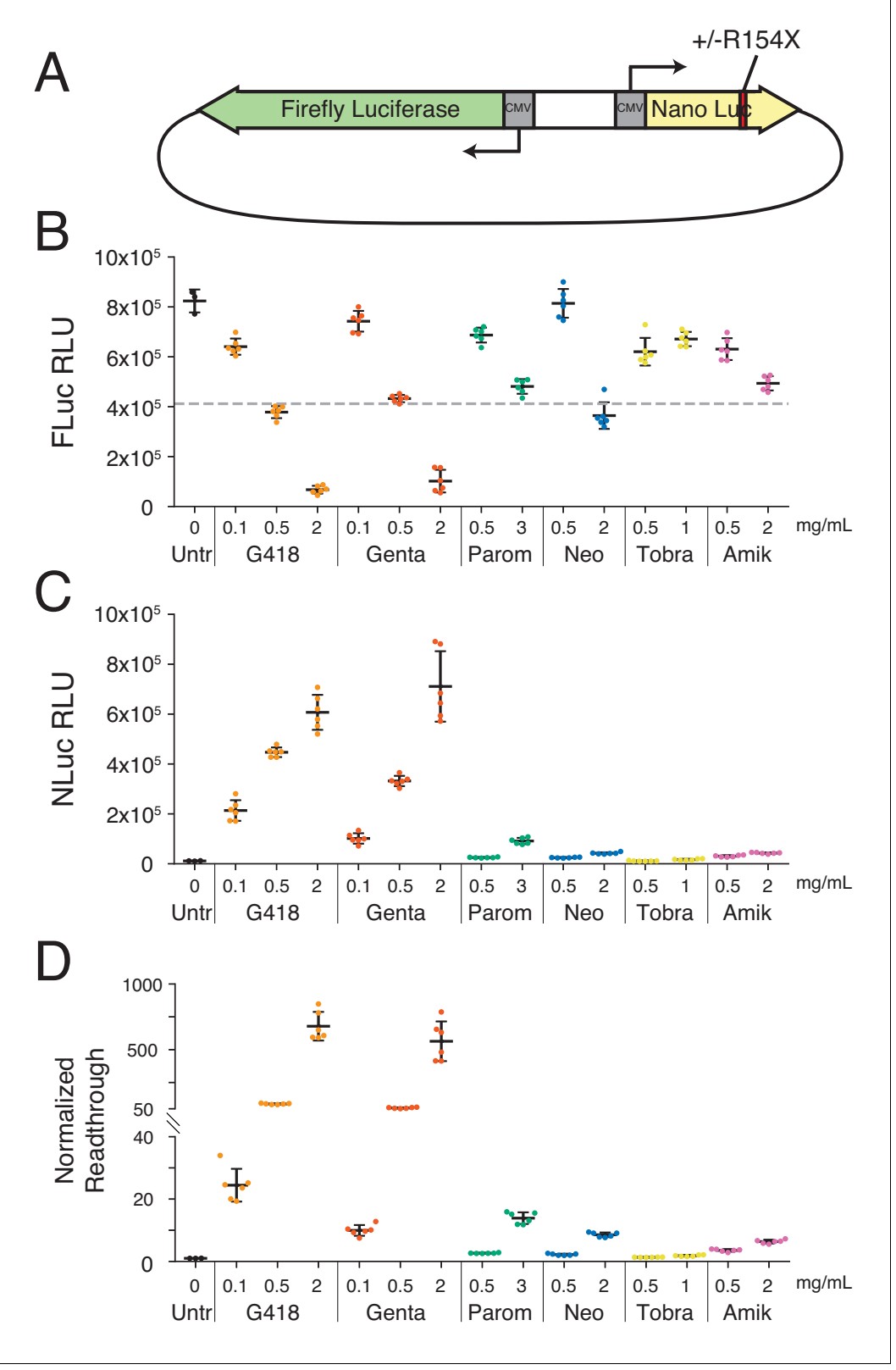

**Figure 1.** Aminoglycosides stimulate readthrough of a PTC in luciferase reporter assays. (**A**) Schematic of reporter constructs used in this study. Full-length firefly luciferase (FLuc) is expressed from the reverse strand and nano luciferase (NLuc) encoding either an arginine, or a UGA stop codon at amino acid 154 is expressed from the forward strand. (**B-D**) HEK293T cells transiently transfected with the PTC-containing reporter after 24 hr of AG

*Figure 1 continued on next page*

*Figure 1 continued*

treatment. (B) Activity of FLuc, (C) or NLuc measured in Relative Light Units (RLUs) as well as (D) normalized readthrough calculated as the NLuc to FLuc ratio for each individual well and normalized to the mean NLuc to FLuc ratio in untreated cells. Mean RLUs or readthrough values are shown with error bars indicating the standard deviations of each measurement. Normalized readthrough in (D) is plotted on two separate axes. Varying concentrations were tested for G418 (0.1, 0.5, 2.0 mg/mL), gentamicin (0.1, 0.5, 2.0 mg/mL), paromomycin (0.5, 3.0 mg/mL), neomycin (0.5, 2.0 mg/mL), tobramycin (0.5, 1.0 mg/mL), and amikacin (0.5, 2.0 mg/mL). Six replicates (separate wells) were collected for each drug treatment condition in this experiment, and at least three separate experiments were performed with all compounds. The dashed line in (B) corresponds to a 50% reduction in FLuc signal relative to untreated cells.

The online version of this article includes the following figure supplement(s) for figure 1:

**Figure supplement 1.** Aminoglycosides tested in this study.

## Ribosome profiling reveals AG stimulation of stop codon readthrough genome-wide

While reporter assays provide a powerful tool for studying readthrough of a given stop codon, the throughput of these assays is inherently limited, their output can be biased by the identify of the amino acid incorporated at a given stop codon (*Xue et al., 2017*), and most importantly, synthetic constructs do not adequately capture termination codons in their endogenous sequence contexts. To address these limitations, we performed ribosome profiling to globally examine readthrough of NTCs in live mammalian cells. By monitoring the presence of ribosomes translating downstream of stop codons, we can identify individual readthrough events independent of protein output. Based on the results from our luciferase reporter, we performed ribosome profiling in HEK293T cells treated with AGs at concentrations that maximized NLuc signal while inhibiting no more than 50% of FLuc signal (*Figure 1B*, dashed-line). Following a 24 hr treatment, cells were lysed and sequencing libraries were prepared for two biological replicates for each treatment condition. Densities of ribosome-protected fragments (RPFs) in coding sequences (CDSs) showed strong correlations between replicates (R > 0.99 for all samples, *Figure 2—figure supplement 1*). Biological replicates were pooled for further analysis to increase read depth for these initial samples.

To measure readthrough of stop codons genome-wide, we performed an average gene (or meta-gene) analysis, aligning all transcripts at their annotated stop codons and calculating normalized ribosome densities in this window. Examining translation in untreated cells (*Figure 2A*, black-line) reveals strong three-nucleotide periodicity in coding regions upstream of stop codons as expected for elongating ribosomes in the CDS. At the stop codon itself, RPFs are enriched (*Figure 2A*, black arrow) as observed in previous studies in multiple organisms, reflecting that termination is slower than elongation (*Ingolia et al., 2011*; *Schuller et al., 2017*). Finally, ribosome density drops precipitously following the stop codon due to the high efficiency of termination, resulting in few ribosomes present in 3′UTRs.

Two immediate differences emerge when comparing global termination in untreated cells to AG-treated cells. First, the peak of terminating ribosomes decreases in cells treated with G418 and paromomycin (*Figure 2A*, orange and green arrows). Second, there is an increase in ribosome density in the 3′UTRs of cells treated with AGs, with the largest increases observed for G418 and paromomycin (*Figure 2B*). The reduction in the peak of ribosomes at stop codons is consistent with increased rates of decoding of NTCs by nc-tRNAs. And, as a corollary, increased rates of readthrough relative to termination at NTCs predict increased density of ribosomes in 3′UTRs. Importantly, the magnitude of the reduction in the peak of terminating ribosomes correlates with the increased density of 3′UTR ribosomes for paromomycin and G418. A 10-minute G418 treatment resulted in equivalent levels of 3′UTR ribosomes as the 24 hr treatment, and treatment with a higher concentration of G418 led to a dose-dependent increase in 3′UTR ribosome density (*Figure 2—figure supplement 2A*). Next, we compared the overall density of ribosomes in a window immediately upstream (−147 to −16 nts, CDS) and downstream (+5 to +100 nts, 3′UTR) of the stop codon to quantify the extent of 3′UTR ribosome enrichment upon AG treatment (*Figure 2C*). At the concentrations tested here, G418 promotes the highest levels of 3′UTR ribosome density (preserving 20% of CDS density), followed by paromomycin, and then gentamicin and neomycin while amikacin and tobramycin only minimally

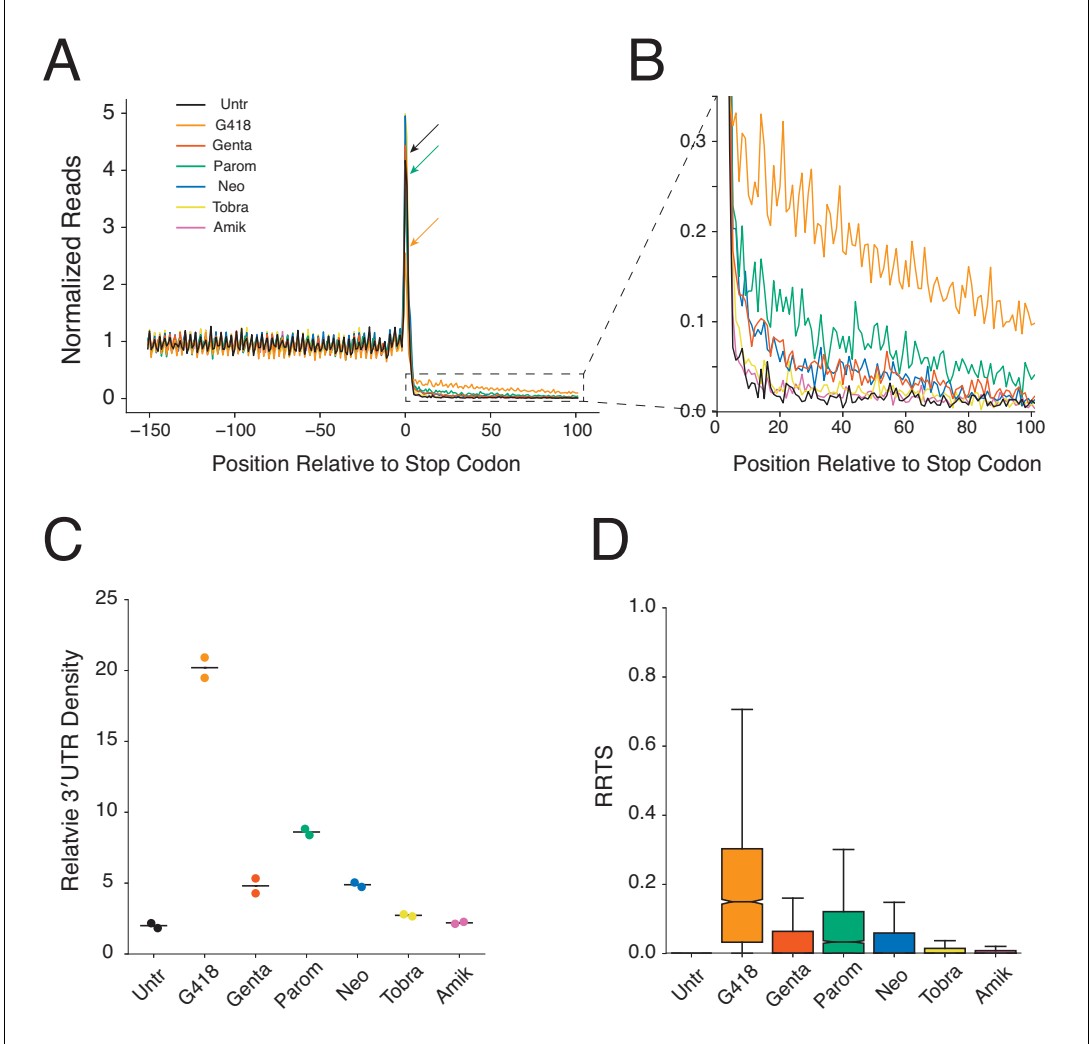

**Figure 2.** Aminoglycosides stimulate genome-wide stop codon readthrough. (**A**) Average gene plot showing normalized ribosome densities relative to the distance, in nucleotides, from the stop codon at position 0. Ribosome densities from untreated cells (black), or cells treated for 24 hr with G418 (orange, 0.5 mg/mL), gentamicin (red, 0.5 mg/mL), paromomycin (green, 3 mg/mL), neomycin (blue, 2 mg/mL), tobramycin (yellow, 1 mg/mL), and amikacin (pink, 2 mg/mL) cells are overlaid. Arrows demonstrate the height of peaks at stop codons for Untr, G418, and paromomycin to facilitate comparison. (**B**) Magnified view of the 3'UTR showing increased densities of ribosomes in this region for AG treated cells. (**C**) Densities of ribosomes in 3'UTRs (from +5 to +100) are plotted relative to densities of ribosomes in the coding sequence (positions −147 to −16) for each AG. Each replicate is displayed, along with the mean value. (**D**) RRTS values are displayed for all genes in pooled replicates using box and whisker plots. Median values are represented with the notch indicating the 95% confidence interval, and whiskers representing 1.5 times the interquartile range. Outliers are not shown. The online version of this article includes the following source data and figure supplement(s) for figure 2:

**Source data 1.** Source data from ribosome profiling analysis used in *Figure 2*.
**Figure supplement 1.** Ribosome profiling sample replicates are well correlated.
**Figure supplement 2.** RRTS values are well correlated among replicates.
**Figure supplement 3.** Aminoglycosides perturb translation initiation and elongation.

promote readthrough above basal levels. Relative levels of readthrough as measured by luciferase activity (*Figure 1D*) generally agreed with 3'UTR ribosome densities for the AGs.

We next evaluated SCR on a per transcript basis. To accomplish this task, we defined a metric referred to here as the **R**ibosome **R**ead**T**hrough **S**core (RRTS, *Figure 2—figure supplement 2B*) where we calculated the density of ribosomes in the region of the 3'UTR (with the assumption that 3'UTR RPFs were generated by SCR) between the NTC and the first in-frame 3'TC, and divided this value by the density of ribosomes in the CDS for every annotated transcript. For each protein-coding gene, only a single transcript isoform possessing the 3'-most termination codon was analyzed to

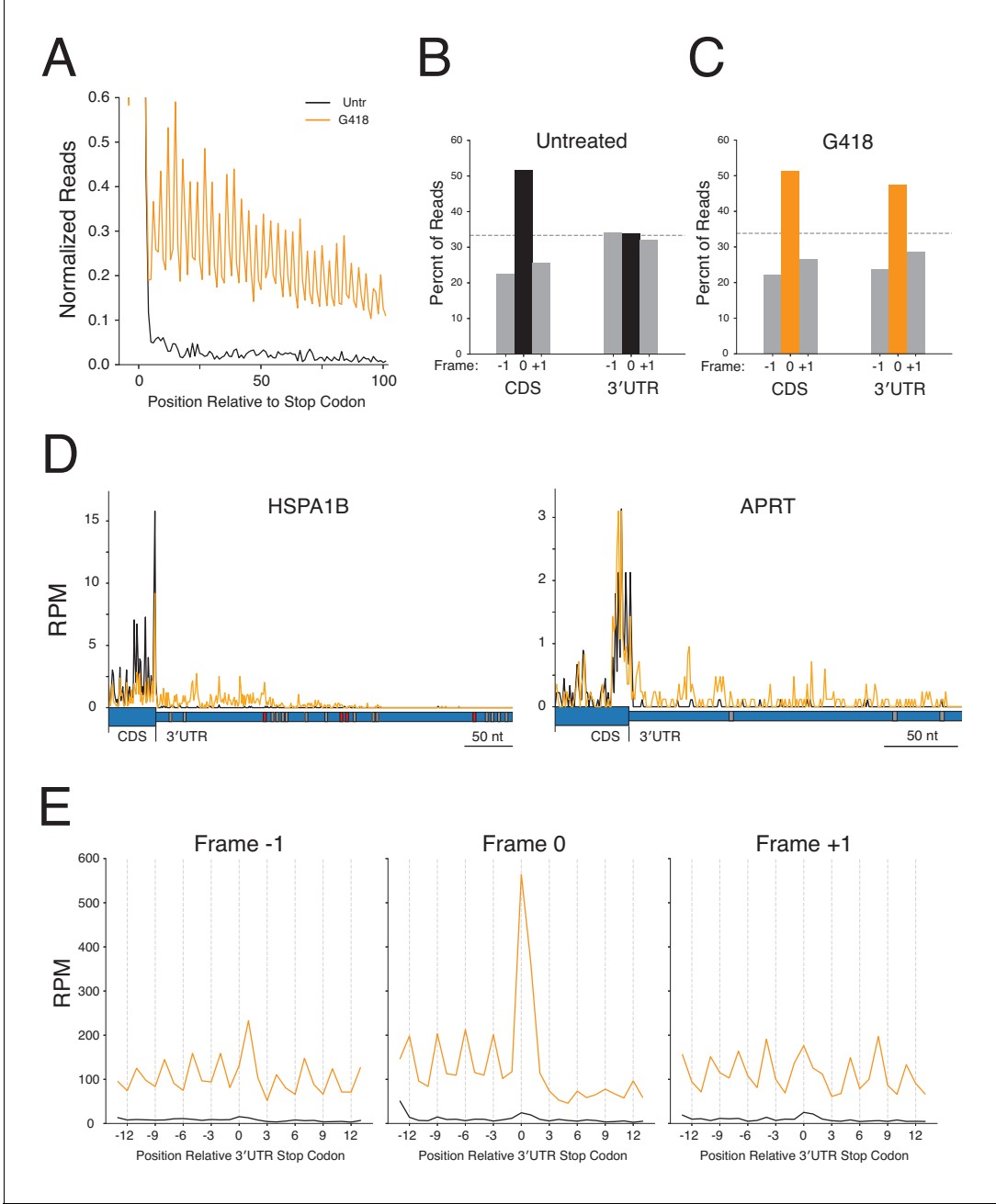

**Figure 3.** 3'UTR ribosomes in G418-treated cells derive from stop codon readthrough. (**A**) Average gene plot showing increased density of ribosomes in 3'UTRs in G418-treated cells (orange) relative to untreated cells (black). Reading frame is analyzed for (**B**) Untreated and (**C**) G418-treated cells showing the percent of ribosomes in a given frame in the CDS and 3'UTR. (**D**) Gene models of HSPA1B (left) and APRT (right) showing translation of the 3'UTRs of these genes. G418-treated cells (orange lines) are overlaid onto untreated cells (black lines). The wider blue bar below the plot indicates the CDS and the narrow blue bar represents the 3'UTR. In-frame 3'TCs are colored in red, while out-of-frame 3'TCs are colored in gray. (**E**) Average gene plots show total ribosome density in the region surrounding the first in-frame 3'TC when found in frame −1 (left), frame 0 (center), or frame +1 (right). Transcripts with additional 3'TCs in this window were excluded for this analysis.

The online version of this article includes the following source data and figure supplement(s) for figure 3:

**Source data 1.** Source data from ribosome profiling analysis used in *Figure 3* and *Figure 4*.
**Figure supplement 1.** Defining features of RPFs in deeper sequencing libraries.
**Figure supplement 2.** Analysis of in-frame stop codons in 3'UTRs genome-wide.

ensure that RPFs aligning downstream of the stop codon reflect translation of the 3′UTR rather than translation of the CDS of an alternatively spliced transcript (see Methods regarding transcript selection). RRTS values were well correlated between biological replicates (R > 0.7, *Figure 2—figure supplement 2C*). As NTC readthrough is normally a rare event, the vast majority of transcripts showed zero RPFs in 3′UTRs of untreated cells in these initial libraries. Treatment with AGs broadly increased RRTS values (*Figure 2D*) indicating that stimulation of NTC readthrough by AGs is a global phenomenon.

In addition to these observed effects of aminoglycoside treatment on translation termination, we also examined effects on translation initiation and elongation. For both of these phases of translation, G418 again proved the most disruptive of the AGs tested here. G418 treatment led to increases in ribosome occupancy at initiation codons (*Figure 2—figure supplement 3A*) while other AGs showed only modest effects on this process. The G418-dependent enrichment of initiating ribosomes was comparable between the 10 min and 24 hr time points, and was increased with a higher concentration of G418. To measure perturbation of translation elongation, we calculated average codon occupancies for all 61 sense codons by comparing the ribosome density at a specified codon relative to the ribosome density of the CDS (*Figure 2—figure supplement 3B*), and averaged these measurements across all occurrences of the given codon. AG treatment globally disrupted codon occupancies to varying extents resulting in a substantial loss-of-correlation between AG-treated and untreated cells. Of note, G418 treatment revealed both amino acid-specific changes wherein codon occupancies on all glycine (orange) and aspartic acid (cyan) codons were increased, as well as codon-specific changes wherein codon occupancy of a single isoleucine (green) codon (AUA) was decreased while occupancy of the other two codons (AUC and AUU) was unaffected. These data reveal that aminoglycosides interfere with every phase of translation and provide additional evidence for the general inhibition of translation observed using luciferase assays (*Figure 1B*).

## Aminoglycoside-induced 3′UTR ribosomes are derived from stop codon readthrough

As termination is normally a very efficient process, ribosomes are rarely found in 3′UTRs. While NTC readthrough events result in translation of 3′UTRs, several additional reactions could also be responsible for the increased 3′UTR ribosome density observed following AG treatment. Previous reports have demonstrated that ribosomes enter 3′UTRs due to readthrough (*Dunn et al., 2013*), frameshifting (*Michel et al., 2012*), failures in ribosome recycling or rescue (*Guydosh and Green, 2014*; *Mills et al., 2016*), or reinitiation of translation (*Young et al., 2015*; *Young et al., 2018*). To establish the primary mechanism responsible for the increased abundance of the ribosomes found in 3′UTRs upon AG treatment, we generated additional deeper ribosome profiling datasets in G418-treated cells as this AG showed the strongest induction of 3′UTR ribosomes. We utilized the increased sequencing depth of these libraries (*Figure 3—figure supplement 1A and B*) to determine whether the ribosomes found in 3′UTRs were derived from readthrough events or other competing pathways.

To verify that sequencing reads aligning in 3′UTRs derive from ribosomes, we compared read length distributions in 3′UTRs and CDSs. Ribosomes in mammalian cells protect, on average, 29–31 nucleotide fragments (*Ingolia et al., 2011*; *Wolin and Walter, 1988*) of RNA while other species, such as RNA binding proteins (*Ji et al., 2016*), protect more variable fragment sizes. When we compare the sizes of 3′UTR and coding sequence reads, we see strong agreement in the distribution of fragment lengths consistent with the argument that 3′UTR reads derive from ribosomes (*Figure 3—figure supplement 1B*).

A useful diagnostic for validating *bona fide* translation in ribosome profiling data is the presence of three-nucleotide periodicity, a signature of elongating ribosomes. Examination of a metagene plot of 3′UTRs in the deep ribosome profiling libraries reveals strong three-nucleotide periodicity in G418-treated cells, but not in untreated cells (*Figure 3A*). By mapping ribosomal A sites of RPFs to single-nucleotide positions, we next calculated the proportions of ribosomes translating in each of the three possible reading frames. As anticipated, both untreated and G418-treated cells show strong enrichment of ribosomes translating in the frame (Frame 0) of the CDS. Strikingly, while the reading frame of the CDS is completely lost in untreated cells (*Figure 3B* - equal representation of RPFs in all three frames), we see strong conservation of frame in the 3′UTRs of G418-treated cells (*Figure 3C*). As the alternative processes that might be responsible for generating 3′UTR ribosomes

(including frameshifting, recycling failure, and reinitiation) should not result in reading frame maintenance, these data strongly indicate that G418 increases 3′UTR ribosome density by stimulating readthrough of NTCs.

Ribosomes that read through the NTC and continue translation are predicted to encounter subsequent in-frame stop codons in 3′UTRs at some frequency. Indeed, over 90% of mRNAs have an in-frame stop codon found in the 3′UTR (*Figure 3—figure supplement 2A*). Given the presence of these in-frame stop codons, ribosomes that read through the NTC will once again face a decision of whether to terminate translation, frameshift, or read through the downstream stop codon. As termination remains the predominant reaction at stop codons, even when cells are treated with G418, ribosome density is predicted to significantly decrease downstream of any in-frame stop codon. For example, examination of HSPA1B (*Figure 3D*) in cells treated with G418 reveals that 3′UTR ribosome density for this transcript remains relatively stable between the NTC and the first in-frame 3′TC, but ribosome density substantially drops after every in-frame 3′TC (*Figure 3D*, red boxes) and not at out-of-frame stop codons (*Figure 3D*, gray boxes). A gene containing no in-frame 3′TCs, such as APRT (*Figure 3D*, right), shows ribosome density throughout the entire 3′UTR as well as ribosomes that reach the poly A tail (data not shown).

We next analyzed termination globally at the first stop codon encountered in 3′UTRs as a function of reading frame (*Figure 3E*). In untreated cells, small peaks of ribosomes are present at the stop codons in all three reading frames suggesting that low levels of translation normally occur in all frames of 3′UTRs (black traces). In contrast, when cells are treated with G418, translation proceeds primarily in the same reading frame as the coding sequence. Upon reaching the in-frame (Frame-0) 3′TC, ribosomes are enriched at the stop codon, followed by a major depletion of downstream ribosome density, similar to what we observe at NTCs (*Figure 2A*). For out-of-frame 3′TCs (−1 and +1 frames) ribosome density is maintained downstream of stop codons indicating that most ribosomes are not translating in these frames.

Given that a majority of ribosomes reaching in-frame stop codons terminate translation, we next asked whether the gradual decline of ribosome density observed in *Figure 3A* can be explained by the fraction of transcripts that have encountered in-frame stop codons. As distance from the NTC increases, so does the probability of encountering a termination codon (*Figure 3—figure supplement 2B*). Satisfyingly, this trend mirrors that of the global decline in ribosome density (*Figure 3A*) as distance from the stop codon increases. Taken together, we conclude that G418-induced 3′UTR ribosomes derive primarily from SCR events.

## Stop codon identity and mRNA context influence probability of readthrough

Numerous reports have demonstrated that the identity of the stop codon influences the likelihood of stop codon readthrough using reporter assays across diverse systems (*Schueren and Thoms, 2016*). To systematically compare the influence of stop codon identity on SCR, we inserted each possible PTC (UAA, UAG and UGA) at six additional positions in our NLuc reporter (W12X, V40X, E51X, H88X, G113X, W134X, and R154X). Importantly, these inserted stop codons were distributed relatively equally along the length of the mRNA and only inserted at positions not predicted to disrupt protein secondary structures (*Lovell et al., 2016*) and thus luciferase reporter activity.

Cells were transfected with these reporters and treated or not treated with G418 to evaluate readthrough at each stop codon. Readthrough efficiencies varied considerably between stop codon positions in a manner that did not appear to be a function of PTC proximity to the poly(A) tail (*Figure 4A* with G418, *Figure 4—figure supplement 1A* without G418). Consistent with previous findings, a general trend emerged that for a particular position, UAA stop codons were least likely to be read through while UGA stop codons were most likely to be read through (readthrough likelihood: UGA > UAG > UAA). It is possible that the variability in SCR between the different PTCs is a consequence of different sequence contexts surrounding each stop codon.

To query readthrough as a function of stop codon identity genome-wide, we again utilized the deeper ribosome profiling data sets. We sorted transcripts by stop codon identity and measured RRTS values for all transcripts in the absence and presence of G418 (*Figure 4B*). While in general, these data provide support that readthrough of NTCs is a rare event under normal conditions regardless of the identity of the stop codon, we did find that UAA and UGA stop codons were significantly more likely to be readthrough than UAG stop codons (p=1.62×10$^{-3}$ and p=4.84×10$^{-7}$,

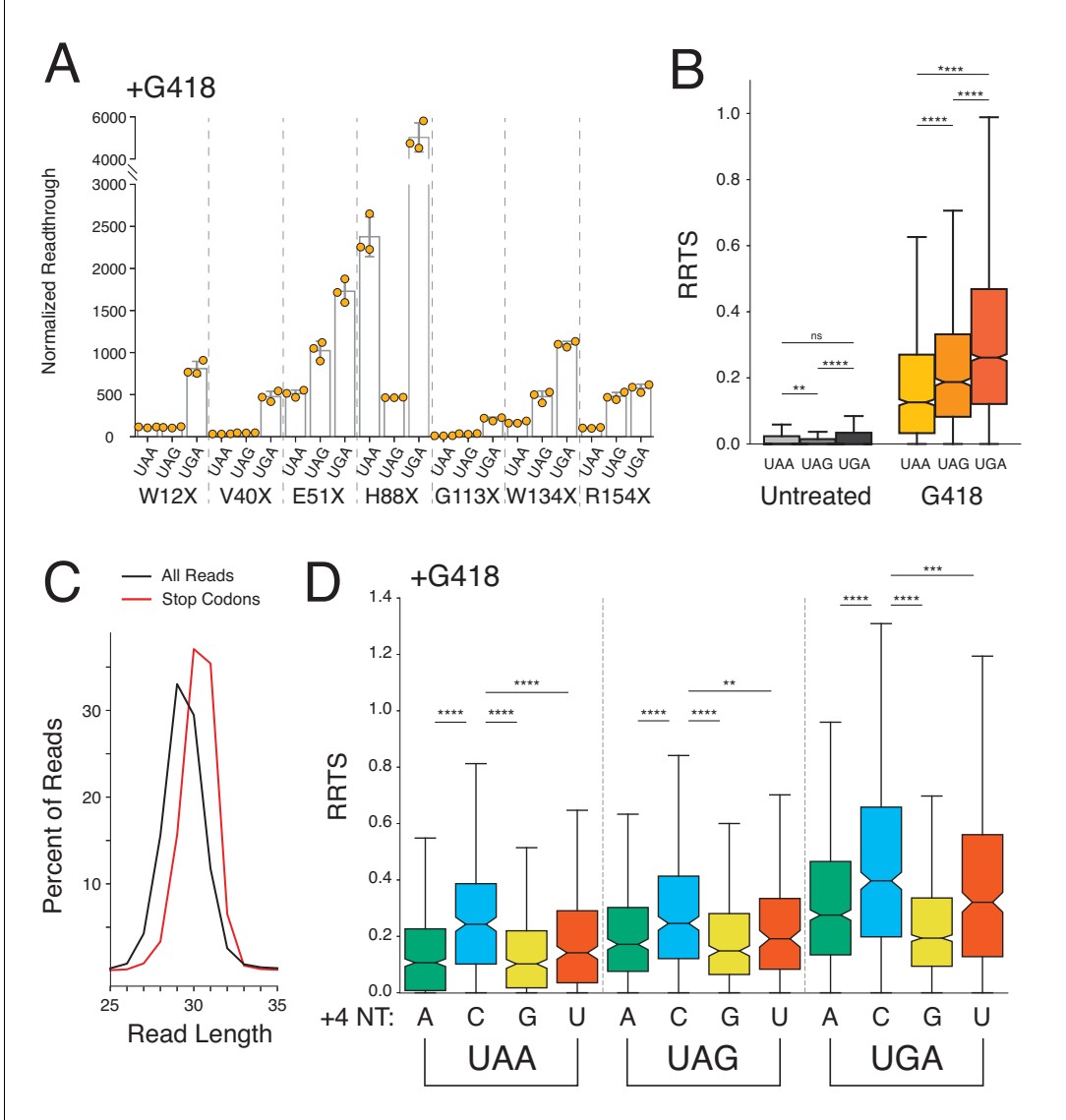

**Figure 4.** Stop codon identity influences readthrough probability genome-wide. (**A**) Luciferase activity in HEK293T cells treated with 0.5 mg/mL G418 for 24 hr was measured for seven different PTCs (W12X, V40X, E51X, H88X, G113X, W134X, and R154X) in NLuc testing all three possible stop codons at each position. Normalized readthrough values - NLuc/FLuc ratios normalized to the lowest NLuc/FLuc ratio in untreated cells (*Figure 4—figure supplement 1A*, V40X-UAG), are plotted for each stop codon. Experiments were performed in triplicate with error bars representing one standard deviation. (**B**) Box and whisker plot showing RRTS values of transcripts sorted by NTC identity. Two-sided Mann-Whitney *U* tests were performed to test for significant differences between groups of transcripts. (**C**) Read size distributions comparing lengths of reads at stop codons (red) to all reads (black) in untreated cells. (**D**) Distribution of RRTS values, in G418-treated cells, comparing the effect of the 4 nt termination codon on readthrough. Within each 3 nt stop codon, presence of a C at the +4 position significantly increases RRTS values (One-sided Mann-Whitney *U* test, p<0.01 for all comparisons). Outliers are not shown.

The online version of this article includes the following figure supplement(s) for figure 4:

**Figure supplement 1.** Stop codon identity influences readthrough in untreated cells.

Mann-Whitney *U*) in untreated cells. More importantly, in G418-treated cells, we identify the same trend as in the reporter assays (UGA > UAG > UAA) but with strong significance as determined by our genome-wide analysis.

Numerous reports have identified an influence of the nucleotide immediately following the stop codon (+4 position) on SCR using reporter assays. Based on structural studies revealing compaction of the mRNA in the A site of the ribosome during termination (*Brown et al., 2015*), stop codons

may be more appropriately considered as four-nucleotide (4 nt) rather than three-nucleotide (3 nt) signals as simply derived from the codon table for tRNA decoding. Indeed, as previously reported (*Ingolia et al., 2011*), measurement of read-length distributions of ribosome protected fragments (RPFs) at stop codons reveal that ribosome profiling accurately captures the protection of an additional nucleotide of mRNA by the ribosome at stop codons (*Figure 4C*). Next, we compared the effects of the twelve possible 4 nt stop codon signals on NTC readthrough (*Figure 4D*) for cells treated with G418. Generally, a purine at the +4 position decreases likelihood of NTC readthrough, likely due to stabilization of the eRF1-mRNA interaction through base stacking between the mRNA and rRNA (*Brown et al., 2015*). For each individual 3 nt stop codon, the presence of a C in the fourth position significantly increases the likelihood of readthrough relative to all other nucleotides (p<0.01 for all stop codons, Mann-Whitney *U*). And, combining the highest-readthrough three-nucleotide codon (UGA) with the highest-readthrough +4 position (C) results in the highest observed readthrough for UGAC compared to all other 4 nt termination codons in both untreated (*Figure 4—figure supplement 1B*) and G418-treated cells (*Figure 4D*).

In light of the influence of the +4 nucleotide on stop codon readthrough, we next asked whether additional positions surrounding the stop codon might also influence stop codon readthrough. We analyzed the sequence context covered by the footprint of the ribosome, fifteen nucleotides upstream and twelve downstream (−15 to +15) of the termination codon using two different approaches. As a first approach, we weighted each stop codon context by the RRTS for that transcript and performed a one-sided two-sample Student's *t*-test using kpLogo (*Wu and Bartel, 2017*) to test whether a given nucleotide at each position increased or decreased the likelihood of stimulating SCR. *P* values were adjusted using the Benjamini-Hochberg procedure (*Benjamini and Hochberg, 1995*) and plotted using Logomaker (*Tareen and Kinney, 2019*) for untreated (top) and G418-treated (bottom) cells (*Figure 5A* and magnification of G418 data in *Figure 5—figure supplement 1A*). As an alternative approach, we used a linear regression model to calculate regression coefficients for every nucleotide in the sequence window (*Figure 5—figure supplement 1B*), a strategy that has been previously applied to readthrough of luciferase reporters (*Schueren et al., 2014*). These distinct approaches yielded striking agreement on the identity of sequence features that yield increased SCR.

Looking broadly across this defined sequence window, several features emerge that influence SCR. Generally, the presence of A's or U's increase SCR probability while C's and G's decreases SCR probability, especially in 3′UTRs, for both untreated (*Figure 5A*, top) and G418-treated (*Figure 5A*, bottom) cells. In untreated cells, along with the stop codon, the most influential positions are the +4 and +5 positions; in G418-treated cells, the influence of the +4 nt and +5 nt are evident, but are now diminished relative to the stop codon. In agreement with previous observations, we also identified the tobacco mosaic virus readthrough signal CAAUUA as the strongest readthrough promoting hexanucleotide signal in the 3′UTR in G418 treated cells (*Anzalone et al., 2019*; *Namy et al., 2001*; *Skuzeski et al., 1991*).

Considering the influence of C's and G's in promoting efficient translation termination relative to A's and U's, we next examined the nucleotide usage of protein coding transcripts 40 nts upstream and 60 nts downstream of the stop codon (*Figure 5B*). As nucleotide usage in the CDS is constrained by the genetic code, we see that many patterns in this region repeat with three-nucleotide periodicity. Intriguingly, despite the inherent A/U richness of 3′UTRs, C's and G's are enriched in the immediate vicinity of the NTC. As distance from the NTC increases, C's and G's occur less frequently until U's and A's eventually become the predominant nucleotides in 3′UTRs. A notable exception to these trends occurs at the +4 position where G's and A's are enriched while C's and U's are depleted, in agreement with the large influence of these nucleotides on SCR at this position (*Figure 5A*). Taken together, it seems that evolution has finely tuned 3′UTRs to promote efficient translation termination.

We next asked what the likelihood of SCR would be at a different class of termination codon, the collection of first in-frame stop codons in the 3′UTR (3′TCs). While 3′TCs are rarely translated under normal conditions, the pervasive translation of 3′UTRs in G418-treated cells allowed for direct comparison between SCR efficiency at NTCs and 3′TCs. For this comparison, we calculated relative downstream ribosome density in a window 30 nts upstream and downstream of the stop codon for NTCs and the first in-frame 3′TC, excluding transcripts that overlapped additional stop codons (*Figure 5D*, *Figure 5—figure supplement 2*). In comparison to NTCs, which preserved 33%

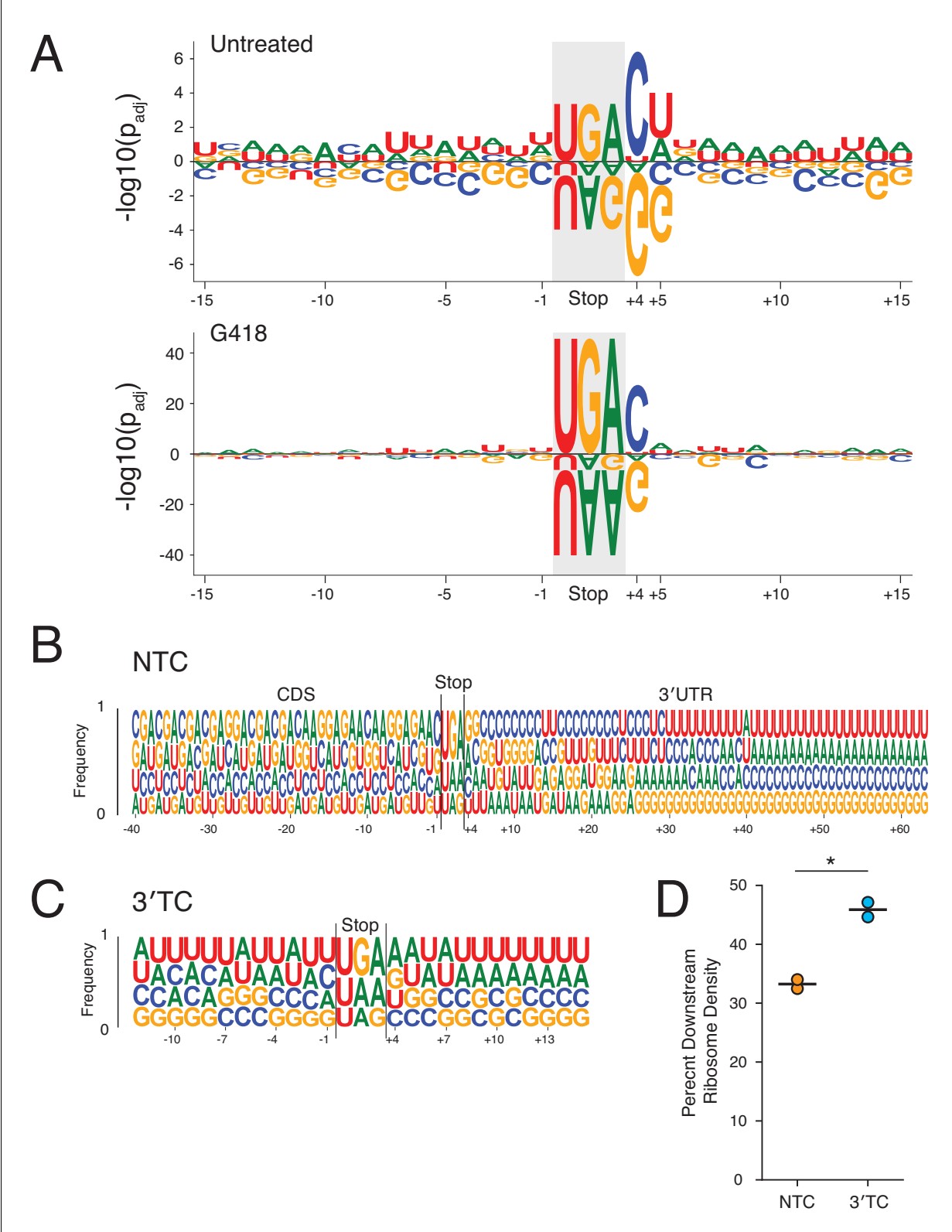

**Figure 5.** Surrounding sequence context influences stop codon readthrough genome-wide. (**A**) Within a sequence window corresponding to the footprint of a translating ribosome at the NTC (15 nt upstream to 12 nt downstream), the likelihood of each nucleotide increasing or decreasing RRTS is plotted with positive values indicating more readthrough and negative values indicating less readthrough. Each nucleotide was tested using one-sided *t*-tests against all other nucleotides at each position for untreated (top) and G418-treated (bottom) cells. *P* values were adjusted using the Benjamini-

*Figure 5 continued on next page*

*Figure 5 continued*

Hochberg correction. Letters are scaled in proportion to the adjusted *P* value. (**B**) The frequencies of each nucleotide are plotted for all positions 40 nt upstream to 60 nt downstream of the stop codon. Nucleotides are plotted in order of increasing frequencies. (**C**) As in (**B**), nucleotide frequencies are plotted for first in-frame 3'TCs, 12 nt upstream to 12 nt downstream of the 3'TC. (**D**) Using normalized ribosome densities in a window 30 nt upstream to 30 nt downstream of stop codons, ribosome density downstream of stop codons was calculated for NTCs and the first in-frame 3'TCs in G418-treated cells. A paired *t*-test was performed revealing significant differences between levels of SCR between 3'TCs and NTCs (p=0.02).

The online version of this article includes the following source data and figure supplement(s) for figure 5:

**Source data 1.** Source data from ribosome profiling analysis used in *Figure 5*.
**Figure supplement 1.** Linear regression analysis of stop codon readthrough.
**Figure supplement 2.** Metagene plots for readthrough at NTCs and 3'TCs.

downstream ribosome density in this window, stop codons in 3'UTRs were more likely to be read through with 46% of the ribosome density maintained downstream of the 3'TC. These observations are consistent with the fact that A's and U's are generally enriched in the stop codon contexts of 3'TCs (*Figure 5C*). It seems likely that purifying selection tightly maintains NTCs but not 3'TCs in mammals (*Belinky et al., 2018*).

## Ribosome profiling reveals readthrough of a premature termination codon

In addition to investigating ribosome readthrough at NTCs and 3'UTR stop codons, we applied ribosome profiling to study readthrough of a natural PTC. For this set of experiments, we used Calu-6 cells that harbor a nonsense mutation (R196X – UGAG) in the tumor suppressor gene TP53. To initially verify that G418 was able to at least partially restore levels of full-length TP53 protein, we analyzed TP53 protein levels by western blotting. In untreated cells, no detectable band was observed for TP53, whereas G418 treatment restored full-length TP53 synthesis in a dose-dependent manner (*Figure 6A*). We also observed production of a second, more prominent band corresponding to truncated TP53. The higher level of truncated TP53 relative to full-length TP53 suggests that even with high levels of G418, termination occurs more frequently than readthrough at the PTC of this mRNA.

We next treated Calu-6 cells with G418 for 24 hr, generated lysates, and sequenced samples using ribosome profiling and RNA-seq. As with HEK293T cells, G418 stimulated global SCR in Calu-6 cells (*Figure 6—figure supplement 1*). Canonical models of EJC-dependent NMD maintain that displacement of EJC's by translating ribosomes protects mRNAs from decay by NMD. Given that TP53 has only a single PTC, we predicted that readthrough of the PTC, and subsequent translation to the NTC, would displace EJCs and protect this message from decay. We measured mRNA levels of TP53 using RNA-seq (*Figure 6B*) and observed a dramatic 16-fold stabilization of TP53 mRNA levels following G418 treatment.

To verify that ribosomes were in fact reading through the PTC, we analyzed ribosome profiling reads on this message for both untreated (*Figure 6C*) and G418-treated (*Figure 6D*) cells. As predicted by low mRNA levels, few RPFs mapped to the TP53 mRNA in untreated cells. Further, all RPFs, save one, were found upstream of the PTC demonstrating little readthrough of the PTC under basal conditions. In contrast, treatment with G418 stimulated readthrough of the PTC, yielding many downstream RPFs, while also enriching reads on the mRNA overall. RPF density upstream of the PTC was greater than that downstream of the PTC in concordance with the western blot analysis (*Figure 6A*). These observations lend support to a model wherein readthrough of a stop codon displaces RNA binding proteins (such as the EJC), protecting the mRNA from NMD (*Keeling et al., 2004*).

## Biological consequences of G418-induced stop codon readthrough

To investigate dysregulation of gene expression by G418, we compared changes in RNA levels (RNA-seq) and translation (ribosome profiling, RPFs) allowing calculation of ribosome occupancies (ROs) by taking the ratio of RPF densities to RNA-seq densities for each transcript (*Figure 7—figure supplement 1*; *Ingolia et al., 2009*; *Xiao et al., 2016*). For the purposes of our discussion, we will treat the RO metric as a reflection of the translational efficiency of a given mRNA. Untreated cells

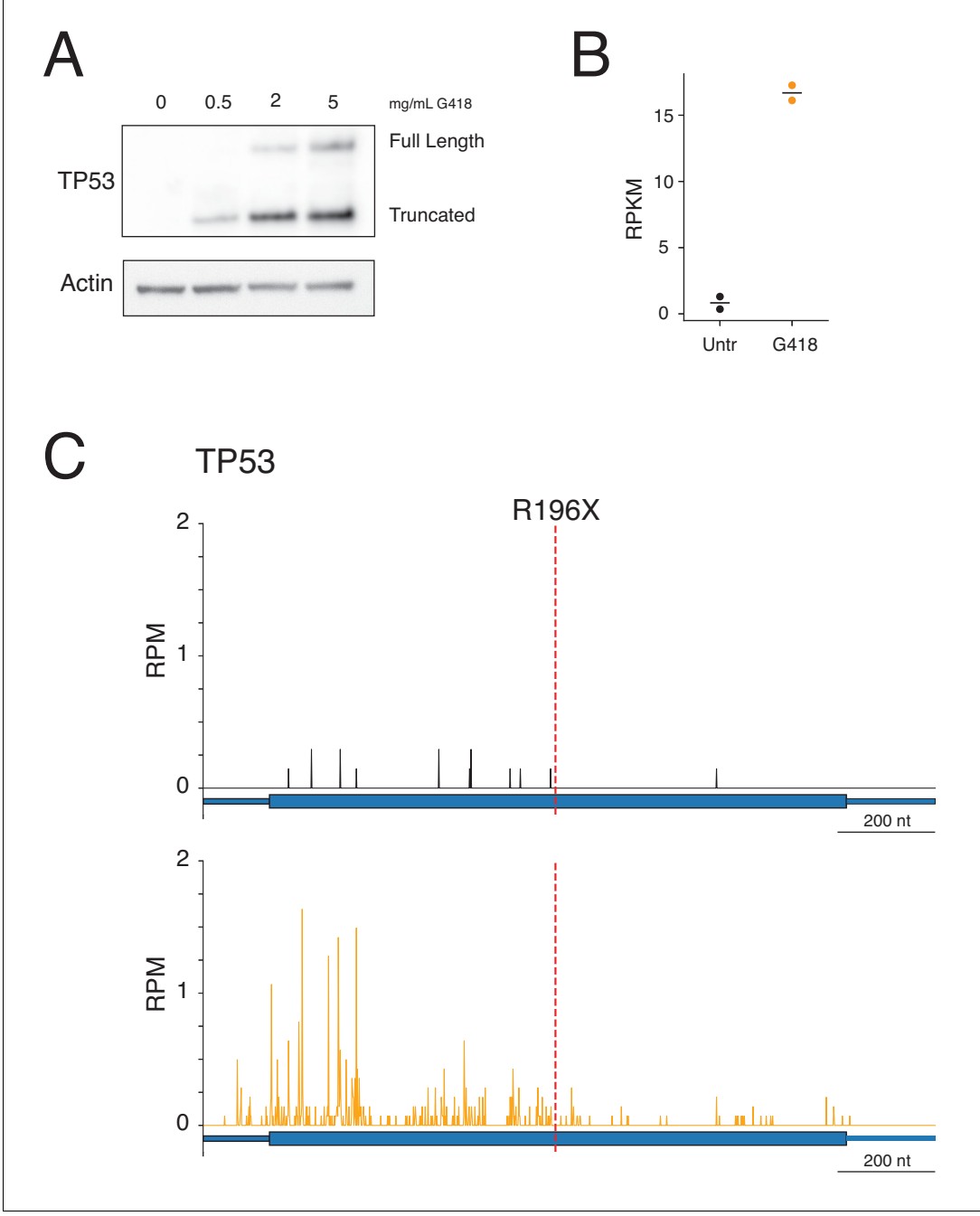

**Figure 6.** G418 stimulates readthrough of a PTC in TP53. (**A**) Western blot showing TP53 protein levels (top) following 24-hour G418 treatment in Calu-6 cells harboring a nonsense mutation in TP53 at R196X. Full-length protein corresponds to the expected size of the readthrough product and the truncated band corresponds to translation termination at the PTC. (**B**) Quantification of mRNA levels in Calu-6 cells by RNA-seq in two biological replicates reveals increased mRNA abundance of TP53 following G418 treatment. (**C-D**) Ribosome profiling reads mapped to TP53 gene model for (**C**) untreated and (**D**) G418-treated cells. The red line at nucleotide 722 of the mRNA indicates the position of the R196X PTC.

The online version of this article includes the following source data and figure supplement(s) for figure 6:

**Source data 1.** Source data from ribosome profiling and RNA-seq analysis used in *Figure 6*.
**Figure supplement 1.** G418 stimulates genome-wide SCR in Calu-6 cells.

were compared to cells treated with G418 at both the 10 min and 24 hr time points (*Figure 7A and B*). We reasoned that a 10 min treatment would capture differences in translation before activation of major transcriptional reprogramming. Indeed, we observe induction of the unfolded protein response (chaperones and foldases) in the levels of both RNA-seq and RPF data at the 24 hr time point (purple dots, upper right quadrant *Figure 7B*) but not at the 10 min time point. These observations are consistent with the fact that the cell must contend with high quantities of misfolded proteins produced by miscoding events as well as C-terminal protein extensions arising from readthrough (*Oishi et al., 2015*). Additionally, we see translational upregulation of ATF4 at the later time point (*Figure 7B*, orange), a critical transcription factor known to be translationally activated during stress (*Vattem and Wek, 2004*).

While the majority of mRNAs are neither dramatically changed in abundance nor differentially translated in response to G418, there are several interesting outliers that we highlight here. First, the histone mRNAs revealed consistent changes in gene expression. For these mRNAs we see that translation is overall decreased and mRNA levels are stabilized, even after only 10 min of G418 treatment (*Figure 7A/B*, red dots). As for the majority of genes, we observe robust readthrough of NTCs on these mRNAs on treatment with G418 (*Figure 7C*). Uniquely, however, we see subsequent translation into a well-characterized conserved hairpin at the terminus of histone 3′UTRs (*Figure 7C*, hairpin highlighted in red).

A second class of genes impacted by G418 treatment is the selenocysteine-containing proteins. Often referred to as the 21$^{st}$ amino acid, selenocysteine tRNA decodes a consensus UGA stop codon as a sense codon, usually in the presence of a downstream SECIS element (*Low and Berry, 1996*). In normal conditions, insertion of selenocysteine appears to be the rate-limiting step for translating many selenoprotein mRNAs, as this position represents the largest peak of ribosomes on these messages in untreated cells as demonstrated for two well translated selenocysteine genes (*Figure 7D*, black lines marked by green arrow). Since G418 potently stimulates readthrough of UGA codons, we wondered whether this AG would alter translation of selenocysteine containing mRNAs. Analyzing expression of selenocysteine genes (*Figure 7A and B*, green dots) revealed that G418 enriches RPFs on several selenocysteine genes relative to mRNA levels. Also, G418 treatment led to a reduction in the peak of ribosomes at UGA selenocysteine codons (*Figure 7D*, orange lines) and increased downstream ribosome density on these mRNAs.

Finally, we noticed a substantial increase in RPFs at both time points along with a decrease in RNA levels at the 24 hr time point for AMD1, a gene whose expression levels are proposed to be regulated by ribosome stalling in both the 5′UTR and 3′UTR. It has been proposed that an upstream open reading frame (uORF) which encodes a specific peptide (MAGDIS*) stalls ribosomes at its termination codon and thus prevents scanning 40S subunits from reaching the authentic start codon of the downstream CDS (*Law et al., 2001*). We find that G418 treatment reduces the peak of ribosomes stalled at the end of the uORF (*Figure 7E*, left) and dramatically increases the RPF density in the CDS at both the early and late time points. A second level of translational regulation was proposed for AMD1 wherein ribosomes that naturally read through the stop codon are trapped on another peptide-regulated stop codon in the 3′UTR (*Yordanova et al., 2018*). It was further proposed that these accumulated ribosomes might form queues that eventually block initiation of the CDS. As previously observed, we identify the peak of ribosomes stalled at the first in-frame 3′TC of AMD1 in untreated cells (*Figure 7E*, right, black arrow). Upon G418 treatment, we see that this peak of ribosomes is strongly enriched (*Figure 7E*, orange arrow), supporting the claim that these stalled ribosomes originate from readthrough of the NTC. We do not, however, observe evidence of queuing ribosomes upstream of this stall site in the 3′UTR, even with the very high level of G418-stimulated readthrough.

## Discussion

Here we used ribosome profiling to examine readthrough of stop codons genome-wide in human cell lines in untreated and aminoglycoside-treated cells. We find that aminoglycosides, and especially G418, generally disrupted the normally accurate process of translation termination leading to high levels of SCR and, in turn, broad perturbation of several cellular processes. Genome-wide levels of NTC readthrough varied considerably between different stop codons allowing investigation into sequence features driving the differences in SCR on the various stop codons. Examination of both 3

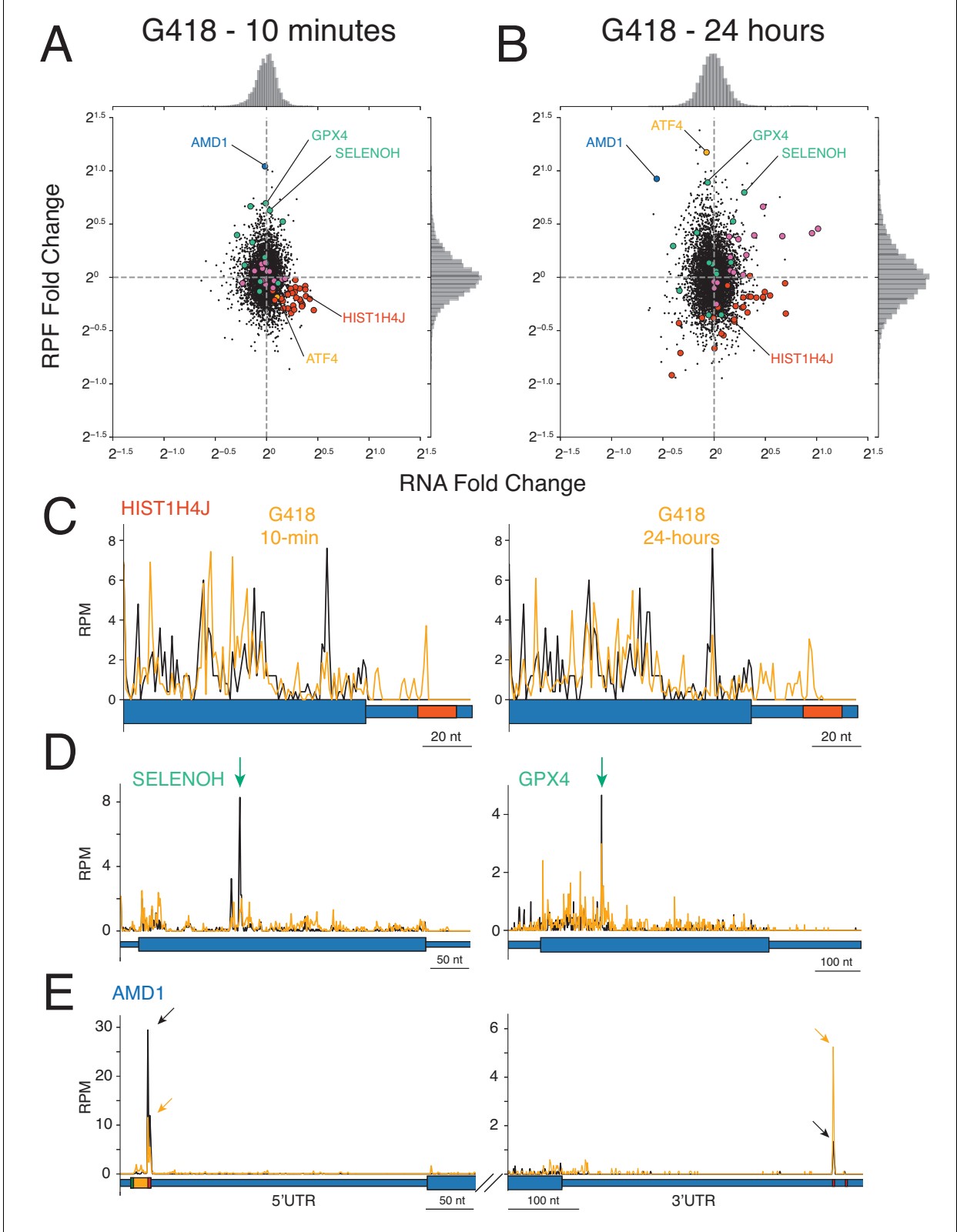

**Figure 7.** Impact of G418 on gene expression. (**A–B**) Changes in mRNA abundance (x-axis) and RPF abundance (y-axis) were calculated for cells (**A**) treated with G418 (0.5 mg/mL) for 10 min or (**B**) treated with G418 (2.0 mg/mL) for 24 hr. Several noteworthy genes are highlighted including histone genes (red), selenoproteins (green), chaperone proteins and foldases (purple), ATF4 (orange), and AMD1 (blue). (**C-E**) Gene models show translation of several genes altered by G418 treatment (orange) relative to untreated (black) cells. Blue bars indicate the position along the mRNA with wide bars

*Figure 7 continued on next page*

*Figure 7 continued*

representing the CDS and narrow bars representing UTRs. (C) Translation of a representative histone gene HIST1H4J at the 10 min (left) and 24 hr (right) time points reveal that G418 induces translation into RNA hairpins of histone mRNAs indicated by the red box in the 3′UTR. (D) Translation of two selenoproteins, SELENOH (left) and GPX4 (right). Green arrows indicate UGA codons normally decoded by selenocysteine tRNAs. (E) Translation of AMD1 is altered by G418 treatment. Reduced peaks of ribosomes stalled in the 5′UTR at the stop codon of the uORF encoding the MAGDIS peptide (left). Arrows indicate the height of peaks at the stop codon for untreated (black) and G418-treated (orange) cells to facilitate comparison. The boxes depicting the start (green) and stop (red) codons and coding sequence (orange) of the uORF are enlarged. G418 treatment increases the peak of ribosomes found in the 3′UTR of AMD1 (right, compare orange arrow to black arrow). No in-frame 3′TCs are present between the NTC and stalling site. The online version of this article includes the following source data and figure supplement(s) for figure 7:

**Source data 1.** Source data from ribosome profiling and RNA-seq analysis used in *Figure 7*.
**Figure supplement 1.** Statistical analysis indicating that G418 alters ribosome occupancy of a subset of genes.

nt and 4 nt termination codons revealed a clear relationship between stop codon identity and SCR in response to G418 treatment; we define genome-wide an increased likelihood of G418-induced SCR with the order UGA > UAG > UAA for the 3 nt stop codons and C > U > A > G for the +4 nt position (*Figure 4B, D*). In addition to the identity of the termination codon, nucleotides more broadly surrounding the NTC also influence likelihood of SCR, especially at the +5 nt position in untreated cells. Generally, the presence of A's or U's near the termination codon increases the probability of SCR while C's and G's favor translation termination (*Figure 5A*). The general observed enrichment of C's and G's following NTCs within the first 20–30 nts of 3′UTRs that we document likely results from evolutionary pressure to terminate translation efficiently (*Figure 5B*). Critically, the first in-frame 3′TCs show higher average levels of SCR in G418-treated cells in comparison to NTCs (*Figure 5D*). This difference likely arises from the enrichment of A's and U's in the stop codon contexts of 3′TCs relative to NTCs, providing further evidence for the important role of stop codon context in regulating translation termination.

We also noted that treatment of cells with G418 changed the relative importance of features that impact SCR. In untreated cells, SCR is strongly driven by the identity of the +4 and +5 nucleotide of the stop codon motif (*Figure 5A*, top). By contrast, in G418-treated cells, SCR is predominantly driven simply by the identity of the stop codon itself (and in particular by UGA) (*Figure 5A*, bottom). We suggest that these striking changes in prediction of readthrough probability between the 3 nt stop codon and the +4 nt contribution could be explained by differences in A site binding mechanisms employed by eRF1 and tRNAs. In untreated cells, where termination by eRF1 is the dominant outcome relative to stop codon readthrough, the sequence context that best stabilizes the eRF1-mRNA interaction effectively predicts the readthrough probability; in particular, the strong contribution of the +4 nucleotide to eRF1 stabilization makes purines (especially G) a strong negative correlate with readthrough and pyrimidines (especially C) a strong positive correlate, as clearly predicted by the structure (*Brown et al., 2015*). On the other hand, in the presence of G418 when miscoding by nc-tRNAs becomes more favorable, the influence of the 3 nt stop codon that is directly recognized by the nc-tRNA now better predicts SCR probability; the cellular tRNA levels may well contribute to the specificity that we observe. Together, these observations suggest that SCR will be impacted by multiple constraints – the strength of interactions between eRF1 and the stop codon context and the likelihood of decoding a particular stop codon in that particular cellular environment.

The high levels of readthrough stimulated by G418 disrupted several biological processes with one of the most substantial consequences being perturbation of histone gene expression resulting in mRNA stabilization and decreased ribosome density (*Figure 7A and B*). Unique among metazoan mRNAs, histones are not poly-adenylated and instead possess a conserved hairpin structure at the 3′ end of the mRNAs. These hairpins are necessary for all stages of the histone mRNA lifecycle, including translation and mRNA decay (*Marzluff and Koreski, 2017*). Due to the massive requirement for histone proteins needed during cell division, exquisite control of protein synthesis and subsequent histone mRNA decay is required as dividing cells transition through the cell cycle. The substantial SCR resulting from G418 treatment results in ribosomes translating into these conserved hairpin structures (*Figure 7C*). This SCR in turn likely disrupted mRNP complexes that normally stimulate translation initiation (*von Moeller et al., 2013*), consistent with the reduced ribosome density that we observe. Furthermore, displacement of 3′hExo/ERI1 (the 3′ to 5′ exonuclease required to initiate

histone mRNA decay [*Slevin et al., 2014*]) by translating ribosomes may similarly explain the observed mRNA stabilization of these transcripts. Resolution of ribosomes stalled at the ends of histone mRNAs may require the eRF1 homolog PELO, which has been shown to rescue ribosomes from 3′UTRs (*Guydosh and Green, 2014*; *Mills et al., 2016*), and specifically, promote decay of histone mRNAs (*Slevin et al., 2014*). Interestingly, deletion of PELO in mouse epidermal stem cells increases ribosome occupancy on these messages (*Liakath-Ali et al., 2018*). More generally, the role of ribosome rescue factors in dealing with genome-wide readthrough events remains an outstanding question.

In contrast to the histones, ribosome density was substantially increased on the selenoprotein mRNAs following G418 treatment. The large peaks of ribosomes normally stalled at the UGA codon sites for selenocysteine insertion were decreased in G418-treated cells; this likely reflects the fact that UGA now becomes recoded by an nc-tRNA instead of the cognate selenocysteine tRNA. Simple replacement of selenocysteine with other amino acids has been shown to decrease function of several of these proteins (*Axley et al., 1991*; *Berry et al., 1991*). Thus, while increased readthrough of UGA codons may generally increase the amount of full-length protein product for these mRNAs, functional levels of selenoproteins may decline with G418 treatment (*Renko et al., 2017*). While selenocysteine insertion is normally specified by an additional RNA sequence element (SECIS) at particular UGA codons, the extent to which the general availability of selenocysteine tRNA (which normally reads UGA in a cognate fashion) contributes to the relatively higher levels of readthrough of UGA(C) stop codons genome-wide remains unclear.

One of the mRNAs exhibiting the largest changes in mRNA level and RPF density on G418 treatment was the enzyme AMD1 which is critical in determining the balance between polyamines and *S*-adenosylmethionine levels in the cell. At the earliest time point, we see AMD1 is the most translationally upregulated mRNA in the transcriptome while at the later time point, translation still remains very high, but mRNA levels decrease substantially (*Figure 7A and B*). Previous studies have highlighted two distinct translational regulatory features in the AMD1 mRNA. First, there is conserved uORF encoding a specific stalling peptide that prevents termination at the uORF and blocks 40S subunits from reaching the downstream ORF (*Law et al., 2001*). We believe that the increased RPF levels that we observe on AMD1 result from increased readthrough of the uORF stop codon that in turn allows scanning ribosomes to find the downstream start site. Second, previous studies showed striking accumulation of 3′UTR ribosomes in AMD1 which the authors suggested resulted from readthrough of the NTC and led to decreased synthesis of AMD1 and decay of the mRNA through a ribosome queuing mechanism (*Yordanova et al., 2018*). The G418-stimulated SCR on AMD1 similarly led to increased levels of 3′UTR ribosomes at the stall site and reduction in mRNA levels. Importantly, however, we see no evidence of ribosome queuing even under conditions of dramatically increased SCR that might support models of translation initiation inhibition. We wonder whether the observed reduction of AMD1 RNA levels that we observe may instead be mediated by the no-go decay pathway. Recent insights into the processes of cellular mRNA surveillance have converged on collided ribosomes as a minimal signal (*Ikeuchi et al., 2019*; *Juszkiewicz et al., 2018*; *Simms et al., 2017*) for triggering no-go decay and ribosome quality control events. In such a model, possibly as few as two readthrough events could trigger quality control thus providing immediate feedback to downregulate AMD1 expression. Further experiments will be required to explore a potential role of ribosome quality control in regulating expression of AMD1.

Despite the potential for interference with specific processes that we have highlighted here, nonsense-suppression therapeutics provide a promising option to target key disease-causing stop codons for selective readthrough. As such, an understanding of the factors that dictate readthrough probability could help guide efforts in achieving specificity for stimulating readthrough of specific termination codons. While G418 successfully induced readthrough of a PTC in TP53 (the most frequently mutated gene in cancers [*Kandoth et al., 2013*]), it simultaneously caused readthrough of the majority of NTCs. Given the generally random nature of nonsense mutations introduced into coding sequences (for example, when associated with genetic disease), stop codon contexts of PTCs are highly variable. Considering that normal stop codons face evolutionary pressure to terminate efficiently, and that we have found that stop codons in 3′UTRs are more likely to be read through than normal stop codons, we predict that most PTCs will exhibit higher levels of SCR than NTCs and thus there may exist a therapeutic 'Goldilocks' window in which restoration of the targeted gene could be achieved without global disruption of translation termination at NTCs. This is especially

true given that only minimal levels of protein restoration may be required to treat certain disorders (*Keeling et al., 2014*). Our data provide strong support for this possibility.

Targeting a central process such as translation without causing deleterious effects remains challenging. General toxicity may result from AGs for multiple reasons including dominant effects from miscoding during elongation or from SCR leading to gain-of-function effects arising from C-terminal extensions (*Freitag et al., 2012*; *Schueren et al., 2014*). Moreover, a therapeutic drawback of AGs, and more generally for all compounds that stimulate miscoding, are the general inhibitory loss of function effects on both translation elongation and termination. Targeting translation termination codons through the use of suppressor tRNAs (*Lueck et al., 2019*) presents an alternative approach to aminoglycosides with potential for fewer off-target effects as suppressor tRNAs should only impact the fidelity of stop codon recognition for roughly one third of stop codons, but not the overall fidelity of translation. Fortunately, cells also possess the capacity to manage some level of miscoding and SCR by degrading readthrough products (*Arribere et al., 2016*). The increased resolution of the role of stop codon context in regulating translation termination presented here will hopefully aid future development of nonsense suppression therapeutics.

## Materials and methods

### Cell culture

HEK293T cells and Calu-6 cells, purchased from ATCC, were cultured in Dulbecco's Modified Eagle Medium with high glucose, L-glutamine, and sodium pyruvate, supplemented with 10% fetal bovine serum, certified endotoxin-free (Thermo Fisher Scientific). Cells were grown in a 37 °C cell culture incubator in the presence of 5% $CO_2$. Antibiotics were not added to the media at any point, with the exception of the AGs tested in the study. Cell lines were purchased directly from ATCC and passaged fewer than 15 times. HEK293T cells tested negative for mycoplasma contamination.

### Aminoglycoside preparation

AGs (Sigma – Gentamicin sulfate, Paromomycin sulfate, Neomycin sulfate, Tobramycin, Amikacin sulfate; Thermo Fisher – G418 sulfate) were prepared by dissolving compounds in either water or 150 mM Tris pH 8.0 to prepare 50 mg/mL stock solutions of each compound. Liquid preparations were also purchased for G418 and Gentamicin (Thermo Fisher Scientific). Activity of compounds was not influenced by preparation methods (data not shown), so compounds prepared in 150 mM Tris pH 8.0 were used for the majority of experiments. Prior to addition of AGs for ribosome profiling and RNA-seq, AGs were added to media and equilibrated in a cell culture incubator to allow stabilization of pH and temperature before addition to cells.

### Plasmids and DNA cloning

All plasmids in this study were generated using the pcDNA5/FRT/TO vector (Thermo Fisher Scientific). The dual expression cassette containing a shared enhancer and minCMV promoters was subcloned from the pBI-CMV1 (Clontech) into the pcDNA5 vector using restriction enzyme digest and Gibson assembly. Nano luciferase (Promega) and Firefly luciferase (a gift from David Bedwell) were inserted into this construct also using Gibson assembly to generate a template expressing full-length versions of both luciferase constructs. PTCs were then inserted via site-directed mutagenesis using the QuikChange Lightning Multi Site-Directed Mutagenesis Kit (Agilent) to generate the constructs tested here.

### Luciferase assays

To measure luciferase activity, HEK293T cells were transfected using Lipofectamine 3000 transfection reagent (Thermo Fisher Scientific) in 96 well plate format. Lipofectamine 3000 (0.225 μL/well) and P3000 reagent (0.2 μL/well) was diluted in serum free media (Opti-MeM with GlutaMAX, Thermo Fisher Scientific), and mixed with plasmid DNA (100 ng/well). Four hours after transfection, AGs were added to each well testing all concentrations in triplicate. After 24 hr of AG treatment, cells were removed from the incubator and equilibrated to room temperature. Luciferase activity was measured using the Nano-Glo Dual-Luciferase Reporter Assay System (Promega). Cells were

lysed through addition of Firefly luciferase reagent by direct injection, followed by measurement of Nano luciferase activity using a Synergy H1 microplate reader (BioTek).

## Western blotting

Cells were lysed in RIPA lysis buffer (25 mM Tris pH 7.6, 150 mM sodium chloride, 1% NP-40, 0.1% SDS, 1% Sodium deoxycholate) supplemented with Roche cOmplete, mini, EDTA-free (1 tablet per 3 mL of lysis buffer), 100 uM PMSF, 1x mammalian protease inhibitor cocktail (Sigma), after brief wash with 1x phosphate buffered saline (Thermo Fisher Scientific). Cells were scraped in the presence of ice-cold lysis buffer and lysed on ice for 10 min. Lysates were then clarified by centrifugation at 21,000 $g$, 4°C, for 10 min and soluble fractions were flash frozen in liquid nitrogen and stored at −80°C. Total protein levels were quantified using a standard curve by BCA Protein Assay (Thermo Fisher Scientific) and absorbance was measured on a Synergy H1 microplate reader (BioTek). 6x SDS loading buffer was added to lysates, samples were denatured at 98°C for 10 min, and 20 µg of protein were loaded to 4–12% Criterion XT Bis-Tris protein gels in 1x XT MES buffer (BioRad). Proteins were then separated using SDS-PAGE at 150V for 1 hr, and transferred to PVDF membrane via Trans-Blot Turbo Transfer System (BioRad). Following blocking with 2.5% milk (Santa Cruz Biotechnology) dissolved in 1x TBS-T for 1 hr at room temperature, the primary antibody to TP53 (Santa Cruz Biotechnology, p53 (DO-1)) diluted 1:100 in 2.5% milk, 1x TBS-T was added and membranes were incubated overnight at 4°C with gentle agitation. Membranes were then washed 3x with 1x TBS-T, incubated with HRP-conjugated secondary antibody (goat-anti-mouse IgG #32430, Thermo Fisher Scientific) diluted 1:1000 for 1 hr at room temperature, and washed again 3x with 1x TBS-T. HRP activity was probed by adding SuperSignal West Femto Maximum Sensitivity Substrate (Thermo Fisher Scientific), and chemiluminescence was detected using a G:BOX (Syngene). Membranes were stripped using OneMinute Advance Western Blot Stripping Buffer (GM Biosciences), reprobed with β-Actin-HRP (Cell Signaling Technologies) diluted 1:5000 for 1 hr at room temperature, and imaged again for detection of luminescent signal.

## Sample preparation for ribosome profiling and RNA-seq

Ribosome profiling samples were prepared as previously described *Ingolia et al. (2012)* with several modifications. AGs were added to cells cultured in 10 cm tissue culture dishes for either 24 hr, or 10 min prior to harvest. Following removal from the incubator, cells were briefly dislodged by scrapping, transferred to 15 mL conical tubes, and pelleted via centrifugation at room temperature for 5 min at 400 g. Cycloheximide was not added to cells at any point prior to lysis. Importantly, we omitted the 1x PBS-wash commonly performed during sample harvesting, as this dramatically decreased the stimulation of SCR otherwise observed for AGs in this assay (data not shown). Media was aspirated and cells were lysed in 200 µL of ice-cold lysis buffer (20 mM Tris-Cl pH 7.4, 150 mM NaCl, 5 mM MgCl$_2$, 1 mM DTT, 100 µg/mL cycloheximide, 1% Triton X-100, 2.5 U/mL Turbo DNase (Thermo Fisher Scientific)) on ice for 10 min. Lysates were clarified via centrifugation at 21,000 g, 4°C, for 10 min. The soluble ribosome-containing supernatant was removed, and separate aliquots were flash-frozen with liquid nitrogen for later quantification, ribosome profiling, and RNA-seq.

## Preparation of ribosome profiling sequencing libraries

Sample lysates were quantified using the Quant-iT RiboGreen RNA Assay (Thermo Fisher Scientific) and fluorescence (excitation 485 nm, emission 530 nm) was measured using a Synergy H1 microplate reader (BioTek). 20 µg of total RNA was diluted to a final volume of 300 uL using polysome buffer (20 mM Tris-Cl pH 7.4, 150 mM NaCl, 5 mM MgCl$_2$, 1 mM DTT, 100 µg/mL cycloheximide) and digested with 750 U of RNase I (Ambion) for generation of ribosome protected fragments. After 1 hr of digestion at 25°C while shaking at 400 RPM on a thermomixer, digestions were stopped with 10 uL of SUPERase*In (Thermo Fisher Scientific). Nuclease-treated lysates were underlaid with 0.9 mL of sucrose dissolved in polysome buffer in 13 × 51 mm polycarbonate ultracentrifuge tubes, centrifuged at 100,000 RPM, 4°C, for 1 hr in a TLA 100.3 rotor using a Beckmann-Coulter Optima MAX ultracentrifuge, and RPFs were extracted using the miRNeasy mini kit (Qiagen).

The extracted RNA was size-selected by denaturing PAGE using a 15% TBE-Urea gel, and fragments corresponding to 15–35 nucleotides (or 26–35 nucleotides for select libraries) were excised. RNA fragments were dephosphorylated using T4 PNK (New England Biolabs), and ligated to a 3′

oligonucleotide adapter containing a unique molecular identifier (UMI) hexanucleotide degenerate sequence using T4 RNA ligase 2 – truncated (New England Biolabs) for 3 hr at 37°C. Ribosomal RNA was depleted using RiboZero (Illumina) omitting the final 50°C incubation step in an effort to reduce contamination of short fragments of rRNA. RNA was then reverse-transcribed to cDNA using Superscript III (Thermo Fisher Scientific) using RT primer with a second four-nucleotide UMI sequence, RNNN. cDNA was then circularized using circLigase (Lucigen) and amplified by PCR using Phusion high-fidelity polymerase (New England Biolabs). PCR-amplified libraries were quantified with a Bioanalyzer 2100 (Agilent) using the High Sensitivity DNA kit and pooled in equimolar ratios. Sequencing was performed using a HiSeq2500 (Illumina) at the Johns Hopkins Institute of Genetic Medicine.

## Preparation of libraries for RNA-seq

Total RNA was extracted from 50 or 100 μL of lysate using 1 mL of TRIzol (Thermo Fisher Scientific). Following precipitation, the RNeasy kit (Qiagen) was used to clean-up RNA prior to library preparation. Starting with 1.0 μg of total RNA, RNA-seq libraries were prepared using the low-throughput TruSeq Stranded Total RNA Library Prep Gold kit (Illumina). As for ribosome profiling, cDNA libraries were quantified using a Bioanalyzer 2100 (Agilent) with the High Sensitivity DNA kit and sequenced with a HiSeq2500 (Illumina).

## Analysis of ribosome profiling data

### Read processing and sequence alignment

Following sequencing of ribosome profiling libraries, we performed several pre-processing steps. Reads were deduplicated, collapsing reads with identical sequences and UMIs, using tally (*Davis et al., 2013*). UMI's were then removed by trimming 4 nucleotides from 5′ ends of reads using seqtk (*Li, 2016*), and 3′ adapter sequences were removed using skewer (*Jiang et al., 2014*). FASTQ files were then aligned to noncoding RNA sequences including rRNA sequences from RefSeq (*Supplementary file 2*) along with annotated noncoding sequences present in GENCODE (version 30) (*Frankish et al., 2019*) (Mt_rRNA, Mt_tRNA, rRNA, miRNA, scRNA, scaRNA, snoRNA, snRNA, sRNA, vaultRNA) using STAR (*Dobin et al., 2013*). Reads mapping to noncoding RNAs were discarded and remaining reads were mapped to the hg38 human genome again using STAR. Mapping of spliced reads was guided by annotations from GENCODE (version 30). Multimapping reads were permitted (*Calviello et al., 2016*) allowing mapping to up to 200 positions, but all alignments flagged as secondary alignments were discarded ensuring only single alignments for each read. All genome-wide analyses presented here were also performed using only uniquely mapping reads, and observations were not substantially different for any figure presented here (data not shown). A notable exception are the histone genes which cannot be uniquely aligned due to similarities between conserved sequences and require multimapping for analysis.

## Selection of single transcripts for each protein coding gene

Following alignment to the hg38 genome, reads were assigned to a custom transcriptome annotation based on the GENCODE (version 30) (*Frankish et al., 2019*) annotations for protein coding genes. Protein coding genes were initially filtered, and all transcripts without completed coding sequences or UTRs were discarded. All protein coding transcripts where then compared, and a single transcript was selected for each gene based on the following procedure for genes with multiple annotated coding transcripts. All transcripts with the 3′ most stop codon were initially selected, ensuring that all 3′UTRs did not overlap with the CDS of alternatively spliced transcripts when quantifying SCR. Genes with multiple transcripts sharing this stop codon were then filtered, first selecting genes with primary APPRIS transcripts (*Rodriguez et al., 2013*), and then alternative APPRIS transcripts, followed by inclusion in the consensus CDS gene set (CCDS) (*Pujar et al., 2018*), and lastly selecting transcripts with the longest coding sequence for genes without APPRIS or CCDS transcripts for this 3′ most stop codon. For genes still containing multiple isoforms, transcripts were finally filtered by choosing transcripts with the shortest 3′UTRs, and lastly selecting those with the shortest 5′UTRs. After selecting a specific transcript, each transcript was checked for overlap with all transcripts of the nearest gene both upstream and downstream on the same chromosome strand as the transcript being queried and transcripts overlapping any nearby transcripts were discarded. Transcripts with UTRs overlapping annotated pseudogenes were also discarded, as these were

observed to bias RRTS calculation for some genes (data not shown). In total, 17,937 transcripts remained after filtering procedures and were used for downstream analyses.

## Assignment of RPFs to single nucleotide positions

The 5′ ends of all aligned reads, between 15 and 40 nucleotides in length, were then assigned to protein coding transcripts without data normalization. These uniquely assigned reads were summed to calculate the total quantity of reads mapped to valid transcripts. Reads were assigned a second time to calculate normalized read counts in reads per million (RPM) by dividing raw counts by millions of mapped reads for every read length.

Mapping to ribosomal A or P sites was next performed by shifting RPF reads from the 5′ end to the position in the center of the A site or P site codon. Density was calculated for all full-length (28–35 nt reads) and empty A site (20–23 nt reads) (*Wu et al., 2019*) ribosomes, using reads mapping to the start codon to calibrate the correct shift of each read length for every sequencing library. These measurements generally agreed with previous reports (*Wu et al., 2019*) corresponding to offsets for full-length reads of {28:[16], 29:[16], 30:[16], 31:[17], 32:[17], 33:[17], 34:[17], 35:[17]} and {20:[16], 21:[16], 22:[17], 23:[17]} for empty A site ribosomes. All analysis was performed using full-length ribosome footprints (28–35 nt).

## Average gene plots

Average gene plots were calculated for regions surrounding the start or stop codons as previously reported (*Schuller et al., 2017*; *Wu et al., 2019*). For each transcript, ribosome density at every position was normalized by the overall density of reads mapping to the coding sequence for that transcript. Genes with features of insufficient length (start codon: 100 nt upstream to 150 nt downstream, stop codon: 150 nt upstream to 100 nt downstream) were discarded. For analysis in *Figure 2C*, 3′UTR density was calculated by dividing normalized 3′UTR (defined as the region from +5 to +100) density by normalized CDS (positions −147 to −16) density and this ratio is displayed as a percent. Regions surrounding the stop codon were excluded to avoid biases due to the large density of ribosomes at this position.

## RRTS calculation

For every transcript RRTS was calculated by dividing the density of ribosomes in the 3′UTR between the NTC and first in-frame 3′TC by the density of ribosomes in the coding sequence. First, transcripts with fewer than 128 mapped reads were discarded to limit noise from lowly expressed genes (*Ingolia et al., 2009*). To calculate CDS densities, the first 18 nucleotides and last 15 nucleotides of the CDS were excluded from analysis to avoid bias from the large peaks at start and stop codons, and mRNA lengths were adjusted accordingly. For calculation for 3′UTR RPF densities between the NTC and 3′TC, transcripts with fewer than 5 codons between the NTC and 3′TC were discarded, and the first 6 nucleotides following the NTC were not included in these calculations. To test for significant differences between RRTS values as a function of stop codon identity, we used the Mann-Whitney $U$ test (*Mann and Whitney, 1947*). To perform log-transformation of RRTSs, scores of zero were assigned to the arbitrarily small value of $2^{-15}$ to facilitate plotting, but discarded from statistical analysis when calculating Pearson correlations of log-transformed data.

## Codon occupancy

First, the position of all 61 sense codons was determined across all annotated transcripts. Then, to calculate average codon occupancies (or pause scores), the ribosome density within the 3 nt window of a queried codon (provided that the codon was not within the first or last two codons of the mRNA) was divided by the average ribosome density for the CDS of each transcript (excluding the first 15 and last 15 nts). This ratio was then calculated for all instances in the transcriptome of a queried codon, and the average of these ratios was determined to be the codon occupancy for that codon.

## Reading frame analysis

The fraction of full length RPFs (28–35 nts) were assigned to the frame of translation in the CDS (frame 0), or the two other two possible frames (frame −1, frame +1). For calculation of reading

frame in *Figure 3B–C*, normalized ribosome densities presented in metagene analysis (*Figure 3A*) were used to compute reading frames for the CDS and 3'UTR. For calculation of reading frame relative to the first 3'TC found in 3'UTRs presented in *Figure 3E*, transcripts were sorted by the frame of the first 3'TC encountered in the 3'UTR. Total reads, normalized by sequencing depth, were summed in the window from 12 nt on either side of the stop codon. Transcripts with additional stop codons in this window were discarded.

### Relative readthrough of 3' termination codons

For comparison between NTCs and first in-frame 3'TCs presented in *Figure 5D*, normalized ribosome densities were again calculated (as in *Figure 2A and B*) in a window spanning 30 nts on either side of each stop codon, for all transcripts except those that (1) lacked any in-frame 3'TCs, (2) the first in-frame 3'TC occurred within the first 30 nts of the 3'UTR, (3) had fewer than 30 nts downstream of the first in-frame 3'TC, or (4) had additional in-frame 3'TCs in the 30 nt window downstream of the 3'TC. Normalized ribosome densities were calculated upstream and downstream of the stop codons, excluding the codon immediately before and after the stop codon (plotted in *Figure 5—figure supplement 2*). The ratio of downstream density divided by upstream density was plotted (*Figure 5D*) and a paired *t*-test was performed testing the difference between two biological replicates.

### Stop codon context weighting by RRTS values

Stop codon contexts in the region spanning the footprint of the ribosome (defined as 15 nts upstream and 12 nts downstream of the stop codon) was retrieved for all transcripts along with the RRTS for that transcript. Any transcripts with 3'UTRs shorter than 12 nts were discarded. For all positions in this sequence window, we used kpLogo (*Wu and Bartel, 2017*), to perform one-sided two-sample Student's *t*-tests to ask whether a given nucleotide increased or decreased RRTS values relative to all other nucleotides at this position. We used the Benjamini-Hochberg procedure (*Benjamini and Hochberg, 1995*) to control the false discovery rate and plotted the *P* values of the nucleotides in the stop codon context using Logomaker (*Tareen and Kinney, 2019*).

### Linear regression modeling

Regression coefficients for each position within the defined window 15 nt upstream and 12 nt downstream of the stop codon were estimated using a regularized least-squares ('ridge') regression (*Hoerl and Kennard, 1970*) of RRTS values with sequence contexts of all annotated transcripts, similar to a previous study using luciferase reporter data (*Schueren et al., 2014*). The sequences surrounding the stop codon were formalized as binary vectors with 27 × 4 positions (A, C, G, or U) and three positions for the stop codon (UAA, UAG, UGA). The absence (0) or presence (1) of each nt was indicated for each sequence. Regression coefficients were calculated using the sklearn.linear_model. Ridge function from the sklearn python package (*Pedregosa et al., 2011*) where X is 111 x *d* (and *d* is the number of transcripts) and y is the vector for RRTS values for each transcript. A value of $10^3$ was used for alpha, as this minimized residual error. Regression coefficients were plotted using Logomaker (*Tareen and Kinney, 2019*).

### Calculation of nucleotide frequencies

Nucleotide frequencies were calculated in a window surrounding the NTC (40 nt upstream to 60 nt downstream) discarding all transcripts with 3'UTRs shorter than 60 nt, and also for the first in-frame 3'TC (12 nt upstream to 12 nt downstream) discarding transcripts with 3'UTRs shorter than 12 nt downstream of the 3'TC in addition to transcripts with other 3'TCs in this window. The fraction of each nucleotide at every position (stop codons were collapsed to a single position) was then calculated and plotted using Logomaker (*Tareen and Kinney, 2019*).

### Differential expression analysis

Differential expression of RNA-seq and ribosome profiling datasets were analyzed using DESeq2 (*Love et al., 2014*) (data not shown). Logarithmic fold changes were estimated using the 'apeglm' package (*Zhu et al., 2019*). For estimation of differences in translation efficiency, we used Xtail (*Xiao et al., 2016*) with default parameters.

## Statistical analysis

All statistical analysis was performed using the SciPy package in python (*Jones et al., 2001*). Statistical details are described in figure legends and available in *Supplementary file 3*: Statistical Test Results. Asterisks in figures denote significance levels of $p > 0.05$ = ns, $p < 0.05$ = *, $p < 0.01$ = **, $p < 0.001$ = ***, and $p < 0.0001$ = ****.

## Data and software availability

Raw sequencing data and count tables for each sample were deposited in the GEO database (GSE138643). Data analysis was performed using custom software written in Python 2.7 and R 3.4.4 available at https://github.com/jrw24/G418_readthrough (*Wangen, 2020*; copy archived at https://github.com/elifesciences-publications/G418_readthrough).

# Acknowledgements

We thank Phil Thomas and Michael Torres for help in working with Calu-6 cells, David Bedwell for providing a Nano Luciferase construct, Boris Zinshteyn for careful reading of the manuscript and help with statistics and data analysis, Colin Wu for extensive help with coding and data analysis, and all members of the Green lab for helpful discussion and guidance throughout the study. We thank David Mohr and the Johns Hopkins Genetic Resources Core Facility for sequencing assistance. This work was supported by the Cystic Fibrosis Foundation (GREEN16G0). JRW was supported by an NIH training grant for a portion of this study (T32 GM007445).

# Additional information

### Competing interests

Rachel Green: Reviewing editor, *eLife*. The other author declares that no competing interests exist.

### Funding

| Funder | Grant reference number | Author |
|---|---|---|
| Cystic Fibrosis Foundation | GREEN16G0 | Jamie R Wangen Rachel Green |
| National Institutes of Health | T32 GM007445 | Jamie R Wangen |

The funders had no role in study design, data collection and interpretation, or the decision to submit the work for publication.

### Author contributions

Jamie R Wangen, Conceptualization, Data curation, Software, Formal analysis, Investigation, Methodology; Rachel Green, Conceptualization, Resources, Supervision, Funding acquisition, Project administration

### Author ORCIDs

Jamie R Wangen (iD) https://orcid.org/0000-0001-7008-5749
Rachel Green (iD) https://orcid.org/0000-0001-9337-2003

### Decision letter and Author response

Decision letter https://doi.org/10.7554/eLife.52611.sa1
Author response https://doi.org/10.7554/eLife.52611.sa2

# Additional files

### Supplementary files

- Supplementary file 1. Key resources table.

- Supplementary file 2. RefSeq Identifiers of sequences used for rRNA depletion.
- Supplementary file 3. Statistical test results.
- Transparent reporting form

## Data availability

Sequencing data have been deposited in GEO under accession number GSE138643.

The following dataset was generated:

| Author(s) | Year | Dataset title | Dataset URL | Database and Identifier |
|---|---|---|---|---|
| Wangen JR, Green R | 2019 | Stop Codon Context Influences Genome-Wide Stimulation of Termination Codon Readthrough by Aminoglycosides | https://www.ncbi.nlm.nih.gov/geo/query/acc.cgi?acc=GSE138643 | NCBI Gene Expression Omnibus, GSE138643 |

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
