## [Decision Letter]

**Acceptance summary:**

The paper describes the investigation of stop codon readthrough in mammalian cells on a genome-wide scale in the absence and presence of aminoglycosides using ribosome profiling. The study shows that aminoglycosides stimulate readthrough of both normal termination codons and premature termination. The study unveils parameters which are important for readthrough efficiencies. In addition, the authors studied the readthrough of downstream 3'UTR stop codons and show that the latter are inefficiently terminated. The study is highly significant for a better understanding of many genetic diseases that are caused by premature termination codons. Most importantly, the study should prompt the search for compounds to augment the readthrough of premature termination codons to restore protein synthesis of the mutated genes.

**Decision letter after peer review:**

Thank you for submitting your article "Stop codon context influences genome-wide stimulation of termination codon readthrough by aminoglycosides" for consideration by *eLife*. Your article has been reviewed by three peer reviewers, and the evaluation has been overseen by a Reviewing Editor and James Manley as the Senior Editor. The following individuals involved in review of your submission have agreed to reveal their identity: David Bedwell (Reviewer #1); Joseph D Puglisi (Reviewer #2); Jonathan S Weissman (Reviewer #3).

The reviewers have discussed the reviews with one another and the Reviewing Editor has drafted this decision to help you prepare a revised submission.

Summary:

The authors demonstrate that aminoglycosides stimulate readthrough of both normal termination codons (NTCs) and premature termination codons (PTCs). Downstream stop codons in 3′UTRs are recognized less efficiently by ribosomes, as compared to NTCs, suggesting that NTCs may have evolved features that resist readthrough. Factors that modulate readthrough include stop codon identity, the nucleotide following the stop codon, and the surrounding mRNA sequence context. They show that different mRNAs, including a histones and stress-related genes, that respond robustly to aminoglycoside treatment, including selenoprotein genes, and S-adenosylmethionine decarboxylase (AMD1).

Essential revisions:

The three reviewers agree that this work is of great interest and merits publication in *eLife*. They offer some thoughtful advice to make amendments and expand the Discussion, in addition to clarifying some text and figures. Please see below the full text of the comments.

Reviewer #1:

The authors analyzed stop codon readthrough in vivo in a genome-wide manner in the absence and presence of aminoglycosides using ribosome profiling. They found that aminoglycosides stimulate readthrough of both normal termination codons (NTCs) and premature termination codons (PTCs) genome-wide. Factors that influenced the efficiency of this process included stop codon identity, the nucleotide following the stop codon, and the surrounding mRNA sequence context. As compared to NTCs, downstream stop codons in 3′UTRs were found to be recognized less efficiently by ribosomes, suggesting that NTCs may have evolved features that resist readthrough. Finally, the authors also found that G418 treatment alters diverse gene expression with particularly striking effects on histone genes, selenoprotein genes, and *S*-adenosylmethionine decarboxylase (AMD1). I think this is an outstanding study that carries out a thorough analysis of stop codon suppression in a genome-wide manner. The analysis is carefully and rigorous, and the results are thoughtfully discussed.

Reviewer #2:

The manuscript of Wangen and Green is a lovely and rigorous study of stop-codon suppression induced by aminoglycoside antibiotics. Such suppression has been known for decades, related to the known miscoding inducing activity for these antibiotics in both prokaryotic and eukaryotic translation. In addition, there is renewed interest in stop codon readthrough agents to treat diseases caused by premature stop codons. As such, this study is both timely, and sorely needed, as to my knowledge there has been no genome wide study of suppression. Here the authors first test a range of aminoglycosides in a standard reporter suppression assay, which showed the known effects of G418 and gentamicin, and established dosages for the subsequent ribosome profiling experiments (allowing readthrough and not inhibiting elongation). The ribosome profiling is of exceptional quality and shows clear readthrough into the 3'UTR upon treatment of drug. The results show that this readthrough is clearly from translating ribosomes (three-nucleotide repeat) within the correct zero frame (not caused by frameshifting). They could then calculate at readthrough score and correlate a range of genome wide mRNA signatures for efficient readthrough, including local sequence context around a stop codon. Using a premature stop codon reporter system, they show efficient readthrough of a premature stop codon in p53, and suppression of NMD pathways. They then study a range of different specific RNAs, including a histones and stress related genes, that respond robustly to aminoglycoside treatment. The results are carefully presented, and not overly interpreted, and represent a large leap forward in our understanding of stop codon suppression. The manuscript as such deserves publication in e*Life* after some modifications to address minor points and presentation issues outlined below.

1) The authors should discuss whether they observe any hints of elongation inhibition in the presence of aminoglycosides. This was not apparent from their data.

2) The construction of the Discussion section should be rethought. The authors dive immediately into aminoglycoside effects on specific classes of mRNAs. This seems too detailed to begin. They may want to consider starting with global conclusions about aminoglycosides-role for 4,6 deoxystreptamines (gent), UGA as a winner, the fact there is readthrough etc. and then present the more specific effects of different mRNA subclasses.

3) Preference here, but I think it would be easier for a reader to have names of aminoglycosides spelled out rather than using abbreviations (feels too colloquial as written).

4) The last sentence of the Abstract is very specific, and it may not be immediately clear to a reader why these genes are interesting with respect to stop codon readthrough.

5) Introduction paragraph three citing Mort et al. should read 11% rather than 10% for accuracy.

6) “As NTC readthrough is normally a rare event, the vast majority of transcripts showed zero RPFs in 3′UTRs of untreated cells in these initial libraries. Treatment with AGs broadly increased RRTS values (Figure 2D) indicating that stimulation of NTC readthrough by AGs is a global phenomenon.” – This statement seems to suggest that a small minority of transcripts showed RRTS values that were > 0 in untreated cells. Of those with non-zero RRTS scores, were these repeatable across replicates or is it just noisy signal? Furthermore, were those transcripts especially susceptible to readthrough upon treatment with AGs?

7) “… we did find that UGA stop codons were significantly more likely to be readthrough than UAG stop codons (*P* = 4.84 x10^-7^, Mann-Whitney *U*) in untreated cells” – In the absence of G418 the authors mention that UGA were more likely to be read through than UAG. Can the same be said of UGA and UAG relative to UAA in untreated cells? If not, why?

8) In Figure 5—figure supplement 1A, it is very difficult to read the identity of the stop codon – can only discern sequence from the colors.

9) In Figure 5A, untreated cells: the significance of the +4 base in readthrough is not surprising given structures from Ramakrishnan and coworkers, but the authors should discuss the significance of the +5 base? It seems to be highlighted by the analysis and significantly stronger than the favorability of A/U from +7 to +15.

Reviewer #3:

The notion that stop codons can in some cases be read through has been appreciated for decades and of course forms the basis of a great deal of classic genetics. There has over the years been much interest in understanding the context of what makes for a robust stop codon and how readthrough is influenced by small molecules. This has been motivated both by its fundamental importance to biology and for therapeutic applications including antibiotics and small molecules that enable preferential read through of disease-causing nonsense mutations (nonsense mutations account for approximately 10% of inherited genetic disorders). Classically these studies have relied on reporter genes but more recently there have been efforts based on ribosome profiling including studies from my own lab on read through in *Drosophila*. The present study, however, goes light years beyond what was done previously using both aminoglycosides to enhance read through at natural stop codons as well as read through at premature termination codons and stop codons in 3'UTRs. The paper is filled with interesting observations; the work is well done, the analysis thoughtful and the presentation clear. I particularly like the three-nucleotide periodicity in the footprints from the 3'UTRs as well as the evidence suggesting evolution has tuned the 3'UTRs to enable efficient translation termination. There is also a bit of a cautionary tale as G418 results in considerable disruption to cell physiology including perturbations to histone gene expression. Overall, I am highly enthusiastic about this paper and recommend publication without delay.

---

## [Author Response]

Essential revisions:The three reviewers agree that this work is of great interest and merits publication in eLife. They offer some thoughtful advice to make amendments and expand the Discussion, in addition to clarifying some text and figures. Please see below the full text of the comments.

We thank the reviewers for their positive feedback and enthusiasm for publication of this manuscript. The suggestions from each reviewer have been included in this revised version of the manuscript that has clarified and strengthened the presentation of the data. Several additions to the manuscript include an additional supplemental figure detailing the effects of aminoglycosides on translation initiation and elongation (Figure 2—figure supplement 3), another subfigure added to Figure 2—figure supplement 2 showing the dose-dependent increase in 3′UTR ribosomes from G418 treatment, and Venn diagrams comparing the reproducibility of readthrough detection in untreated and G418-treated cells (Figure 3—figure supplement 1). We have also reorganized the Discussion as suggested by multiple reviewers.

Reviewer #2:[…]1) The authors should discuss whether they observe any hints of elongation inhibition in the presence of aminoglycosides. This was not apparent from their data.

We did indeed observe effects on codon occupancy in cells treated with aminoglycosides and have added a supplemental figure to illustrate these genome-wide effects on translation elongation. Furthermore, we also observe enrichment of ribosomes at start codons in cells treated with G418. We have added a second additional supplemental figure (Figure 2—figure supplement 3) to illustrate these effects.

2) The construction of the Discussion section should be rethought. The authors dive immediately into aminoglycoside effects on specific classes of mRNAs. This seems too detailed to begin. They may want to consider starting with global conclusions about aminoglycosides-role for 4,6 deoxystreptamines (gent), UGA as a winner, the fact there is readthrough etc. and then present the more specific effects of different mRNA subclasses.

We struggled with this and appreciate the feedback. The Discussion has been restructured to better highlight the role of aminoglycosides on stimulating stop codon readthrough at the outset. The order of the Discussion now features genomic features that influence stop codon readthrough, followed by biological consequences of readthrough (histones, selenoproteins, and AMD1), and finally implications of the findings here for development of nonsense-suppression therapeutics.

3) Preference here, but I think it would be easier for a reader to have names of aminoglycosides spelled out rather than using abbreviations (feels too colloquial as written).

The abbreviations for aminoglycosides have been removed and replaced with the full name of each compound in the text.

4) The last sentence of the Abstract is very specific, and it may not be immediately clear to a reader why these genes are interesting with respect to stop codon readthrough.

The final sentence of the Abstract has been modified to highlight the role of miscoding in disruption of gene expression for these genes.

5) Introduction paragraph three citing Mort et al. should read 11% rather than 10% for accuracy.

This value has been corrected in the text.

6) “As NTC readthrough is normally a rare event, the vast majority of transcripts showed zero RPFs in 3′UTRs of untreated cells in these initial libraries. Treatment with AGs broadly increased RRTS values (Figure 2D) indicating that stimulation of NTC readthrough by AGs is a global phenomenon.” – This statement seems to suggest that a small minority of transcripts showed RRTS values that were > 0 in untreated cells. Of those with non-zero RRTS scores, were these repeatable across replicates or is it just noisy signal? Furthermore, were those transcripts especially susceptible to readthrough upon treatment with AGs?

The rarity of stop codon readthrough in untreated cells presented a challenge for capturing these events using ribosome profiling. With libraries sequenced extremely deeply, we anticipate that most if not all transcripts would show detectable levels of stop codon readthrough. A nonzero RRTS value in untreated cells correlates very strongly with the number of raw sequencing reads assigned to the CDS of a given transcript, and only minimally with the actual readthrough level of that transcript. To address this question, we have added Venn diagrams (Figure 3—figure supplement 1B) showing the number of transcripts with nonzero RRTS values for each sequencing libraries in our deepest datasets analyzed in Figure 3. These show strong agreement between replicates in the ability to detect readthrough on identical transcripts, even in untreated cells. There are more transcripts in replicate 1 that are not detected in replicate 2 because replicate 1 had more uniquely assigned sequencing reads (8,929,987 vs 3,888,992).

7) “… we did find that UGA stop codons were significantly more likely to be readthrough than UAG stop codons (P = 4.84 x10^-7^, Mann-Whitney U) in untreated cells” – In the absence of G418 the authors mention that UGA were more likely to be read through than UAG. Can the same be said of UGA and UAG relative to UAA in untreated cells? If not, why?

We added the comparison between UAA to UAG and the corresponding *P* value (Mann-Whitney U) to the text. While UGA codons had higher average RRTS values than UAA codons, these results were not statistically significant (*P* = 0.0585) as indicated in Figure 4B.

8) In Figure 5—figure supplement 1 panel A, it is very difficult to read the identity of the stop codon – can only discern sequence from the colors.

While the stop codons may be difficult to see here, this figure is a magnified image of Figure 5A which clearly illustrates the position of the stop codons. The purpose of presenting this magnified view is to allow comparison between less influential nucleotides in the surrounding stop codon context.

9) In Figure 5A, untreated cells: the significance of the +4 base in readthrough is not surprising given structures from Ramakrishnan and coworkers, but the authors should discuss the significance of the +5 base? It seems to be highlighted by the analysis and significantly stronger than the favorability of A/U from +7 to +15.

We extended our Discussion of the influence of the +5 nucleotide in untreated cells. We hope that, together with two lines in the results addressing the +5 position data, the importance of this position is now apparent to the reader.